# Isoprene and monoterpene emissions in south east Australia: comparison of a multi-layer canopy model with MEGAN and with atmospheric observations

Kathryn M. Emmerson[1], Martin E.Cope[1], Ian E. Galbally[1], Sunhee Lee[1†], Peter F. Nelson[2]

[1]Climate Research Centre, CSIRO, PMB1, Aspendale, VIC 3195, Australia
[2]Environmental Sciences, Macquarie University, NSW 2109, Australia
† Sadly deceased

*Correspondence to*: Kathryn Emmerson (kathryn.emmerson@csiro.au)

**Abstract.** One of the key challenges in atmospheric chemistry is to reduce the uncertainty of biogenic volatile organic compound (BVOC) emission estimates from vegetation to the atmosphere. In Australia, eucalypt trees are a primary source of biogenic emissions, but their contribution to Australian air sheds is poorly quantified. The Model of Emissions of Gases and Aerosols from Nature (MEGAN) has performed poorly against Australian isoprene and monoterpene observations. Finding reasons for the MEGAN discrepancies and strengthening our understanding of biogenic emissions in this region is our focus. We compare MEGAN to the locally produced Australian Biogenic Canopy and Grass Emissions Model (ABCGEM), to identify the uncertainties associated with the emission estimates, and the data requirements necessary to improve isoprene and monoterpene emissions estimates for the application of MEGAN in Australia. Previously unpublished, ABCGEM is applied as an online biogenic emissions inventory to model BVOCs in the air shed overlaying Sydney, Australia. The two models use the same meteorological inputs and chemical mechanism, but independent inputs of Leaf Area Index (LAI), Plant Functional Type (PFT) and emission factors. We find that LAI, a proxy for leaf biomass, has a small role in spatial, temporal and inter-model biogenic emission variability, particularly in urban areas for ABCGEM. After removing LAI as the source of the differences, we found large differences in the emission activity function for monoterpenes. In MEGAN monoterpenes are partially light dependent, reducing their dependence on temperature. In ABCGEM monoterpenes are not light dependent, meaning they continue to be emitted at high rates during hot summer days, and at night. When the light dependence of monoterpenes is switched off in MEGAN, night time emissions increase by 90 – 100% improving the comparison with observations, suggesting the possibility that monoterpenes emitted from Australian vegetation may not be as light dependent as vegetation globally. Targeted measurements of emissions from in-situ Australian vegetation, particularly of the light dependence issue are critical to improving MEGAN for one of the world's major biogenic emitting regions.

## 1 Introduction

The emission of biogenic volatile organic compounds (BVOCs) by vegetation and their impact on air quality was first noted by Went (1960), who proposed that their oxidation produced the "blue-haze" often seen over forested areas. Subsequent studies of biogenic emissions estimated the quantity and type of chemical species emitted from specific vegetation sources. The two most important BVOCs in terms of emissions are isoprene, and the group of $C_{10}H_{16}$ monoterpene species.

The high reactivity of BVOC emissions has significant impacts on tropospheric chemistry at both regional and global scales. In the presence of light and oxides of nitrogen ($NO_x$), BVOCs undergo a complex series of chemical reactions that can significantly affect atmospheric chemistry by increasing ground level ozone production. The interaction of BVOCs with anthropogenic pollutants (e.g. $NO_X$, $SO_2$, $NH_3$ and organic carbon) can also lead to the production of low volatility organic compounds that can condense to form secondary organic aerosols (SOA) (Hallquist et al., 2009; Xu et al., 2015; Lin et al.,

2013). SOA can affect the radiation budget at the surface of the earth, potentially impacting on climate. Biogenic SOA also contributes to the total atmospheric fine particle burden and exposure to these particles can have deleterious impacts on human health (Schwartz et al., 1996).

BVOC emissions have been studied extensively, however significant uncertainties remain in their estimation. These uncertainties include both variability in the vegetation types and variability in the emission rate. Emission rates depend on many parameters including sunlight, temperature and water availability. One of the most commonly used algorithms for estimating BVOC emission rates was proposed by Guenther et al. (1991; 1993; 1995; 1997) providing the basis for the Model of Emissions of Gases and Aerosols from Nature, MEGAN, (Guenther et al., 2006; Guenther et al., 2012). MEGAN has been used to estimate the BVOC emissions within many atmospheric chemistry models (Heald et al., 2008; Pfister et al., 2008; Stavrakou et al., 2009; Emmons et al., 2010; Millet et al., 2010; Situ et al., 2013; Kim et al., 2014; Stavrakou et al., 2014; Tilmes et al., 2015).

The south east coastal ecosystem of Australia is dominated by eucalypt trees, and is identified as a global BVOC emitting hotspot (Guenther et al., 2006). However recent work by Emmerson et al. (2016) demonstrated considerable discrepancies using MEGAN when compared to atmospheric observations over south-eastern Australia. Emmerson et al. (2016) postulated that the discrepancies were due to unrepresentative emission factors, the majority coming from studies both in Australia and overseas on eucalypt saplings under laboratory conditions. The VOC emissions from Australian vegetation may be different in magnitude and behaviour from those studied in the northern temperate regions and in the tropics because Australian vegetation was isolated from other regions for many tens of millions of years and in general adapted to infertile deeply weathered ancient soils and a regime of intense fires (Orians and Milewski, 2007), factors that could affect the evolutionary biology of plant VOC emissions (Fernández-Martínez et al., 2017). These questions on VOC synthesis are beyond the scope of this paper. Simpler causes of model-observation mismatch are explored first.

Here we use further modelling and comparisons with atmospheric observations to try to understand why MEGAN performs poorly over south-eastern Australia. A comparison of MEGAN with the unpublished locally developed Australian Biogenic Canopy and Grass Emissions Model (ABCGEM) could provide useful scientific insights. South East Australia is a region with very few experimental studies of BVOCs, and comparison with ABCGEM results may be an efficient way to identify the limitations and strengths of MEGAN here. ABCGEM is much simpler than MEGAN. In comparing the two models, the original surface vegetation descriptions and emission factors used by each model have been maintained, enabling us to calculate a total uncertainty in biogenic emissions for the Sydney Greater Metropolitan Region (GMR). We need to understand these inputs, both temporally and spatially, as they influence the model results. We also test ABCGEM using the input leaf area index (LAI) dataset used by MEGAN.

This paper is arranged as follows: section 2 describes the observations used in the study and includes two previously unpublished datasets in the GMR. We then introduce ABCGEM and the emission factors used. Section 2.3 documents how ABCGEM and MEGAN are set-up within the CSIRO-CTM (C-CTM, Cope et al. (2004)). The results of the emission flux and modelled volume mixing ratio comparison are presented in section 3 together with discussion on the causes of the differences. The conclusions in section 4 bring together our current experience with Australian BVOC modelling, and recommend further work to improve isoprene and monoterpene emission estimates in the region.

## 2   Methods

### 2.1 Details of campaign atmospheric BVOC measurements

Figure 1 shows the locations of the five field campaigns conducted within the Sydney GMR, The Sydney Particle Studies SPS1 and SPS2, Measurements of Urban Marine and Biogenic Air (MUMBA), and campaigns at Bringelly and Randwick. Each campaign measured hourly concentrations of isoprene and monoterpenes using the same PTR-MS instrument and employed standard calibration gases. Observations of monoterpenes by PTR-MS are based on the calibration and measurement of the combined monoterpene species at mass to charge ratio m/z = 81 for the Bringelly and Randwick campaigns and at mass to charge ratio m/z = 137 for the later SPS1, SPS2 and MUMBA campaigns. The change was made to improve sensitivity and reduce potential interferences. Three of the campaigns were documented in Emmerson et al. (2016): SPS1 and SPS2 were located at Westmead, a suburban site 21 km west of Sydney (150.9961°E, 33.8014°S). SPS1 ran from 18 February – 7 March 2011, and SPS2 from 14 April – 14 May 2012, (Cope et al., 2014). The Westmead site is located next to a grass playing field within hospital grounds, with a line of trees to the west and south, separating the site from trains, roads and housing beyond. The MODIS LAI value for Westmead is 1.2 $m^2$ $m^{-2}$. Dunne et al. (2018) have shown night time interference from wood smoke compounds in the isoprene signal taken during SPS2. Therefore the SPS2 isoprene observational dataset is restricted to daylight hours between 9am and 6pm. MUMBA was situated near the coast at Wollongong, (150.8995°E, 34.3972°S) from 22 December 2012 – 15 February 2013 (Paton-Walsh et al., 2017). The MUMBA site is also grassy (LAI of 1.7 $m^2$ $m^{-2}$), separated from the ocean 0.5 km to the east by a strip of eucalypt trees. A 400 m eucalypt forested escarpment is 3 km to the west.

A suite of meteorological data, including wind speed and direction were taken at each of the field campaign sites, with details given in the indicated literature. Polar bivariate plots are also shown in Figure 1 which give observed isoprene volume mixing ratios by wind speed and direction at each of the campaign sites. These show that the peak isoprene measurements are not always associated with the dominant wind directions, but are correlated with the directions of the forested regions to the northwest and west of each of the sites.

### 2.1.1 Bringelly and Randwick

PTR-MS observations were undertaken in summer 2007 at Bringelly, a semi-rural site (150.7619°E, 33.9177°S, 24 January – 27 February 2007), and Randwick, 8 km from Sydney centre (151.2428°E, 33.9318°S, 28 February – 19 March 2007). Both sites are air quality management stations and take wind speed and direction, temperature and relative humidity measurements, along with ozone, NOx and particulate matter (www.environment.nsw.gov.au/AQMS/SiteSyd.htm). The inlet height for the PTR-MS instrument was approximately 4.5 m at both sites. Bringelly is located on reserve of open grassed council land (LAI of 2.1 $m^2$ $m^{-2}$), with occasional trees and bordered by Ramsay road at 53 m elevation. Low density housing is to the east. The heavily eucalypt-forested Blue Mountains are 16 km to the west, which is where the source of the observed isoprene comes from. However the predominant wind directions are from the south-west and east.

The Randwick station at 28 m elevation is sited on a grassland paddock within army barracks, bordered by trees. The barracks are within a housing suburb (LAI of 0.5 $m^2$ $m^{-2}$). The dominant wind direction is from the south, with the dominant BVOC source coming from the north-west, consistent with the SPS1 BVOC source direction.

### 2.2 The Australian Biogenic Canopy and Grass Emissions Model (ABCGEM)

The ABCGEM model was developed 15 years ago at CSIRO to provide a spatially and temporally resolved interactive biogenic emission inventory for the C-CTM (Cope et al., 2004). ABCGEM treats the emissions of BVOCs from a 10-layer tree canopy, for which in-canopy gradients of temperature and radiation are parameterised. The approach is largely based on the light and temperature algorithms of Guenther et al. (1993) and Guenther (1997), and is documented in the supplementary material. ABCGEM uses LAI to calculate the column biomass, $B_m$, and fractional area taken up by vegetation in each grid cell, and to scale the leaf level emission rates. ABCGEM also accounts for grass emissions (see technical report by Cope et al. (2009)),

however as leaf level eucalypt emission rates are 1000 times higher than grass in the Sydney air shed, the grass module will not be discussed here.

### 2.2.1   Choice of ABCGEM emission factors

We take measured leaf level emission rates and convert them into landscape emission factors for eucalypts by scaling with the column biomass of each grid cell (per unit ground area), making them a function of the LAI (see supplementary material equation 3). In ABCGEM the leaf-level isoprene emission rate for trees is 25 $\mu$g-C $g^{-1}$ $h^{-1}$, representing the average isoprene emission rate for measurements conducted on Eucalypt and Casuarina species (He et al., 2000; Benjamin et al., 1996; Nunes and Pio, 2001). The normalised lumped monoterpene emission rate for trees is 2.5 $\mu$g-C $g^{-1}$ $h^{-1}$, based on measurements on Eucalypt, Callistemon, and Pittosporum species (He et al., 2000; Benjamin et al., 1996; Nunes and Pio, 2001). The ABCGEM emission factors are compared with those from MEGAN for the Sydney domain in the results section.

### 2.3  The CSIRO Chemical Transport Model

The C-CTM is a coupled, three-dimensional Eulerian chemical-transport modelling framework, used to generate spatial and temporal fields of gas and aerosol phase species (Cope et al., 2004). The framework consists of modules to predict the meteorology, emissions, chemical processing and wet and dry deposition. An 80km resolution Australia-wide domain houses three successively smaller modelling domains nested at 27km, 9km and 3km resolution respectively. The highest resolution inner grid is centred on either Westmead or Wollongong and extends for 180km north-south and east-west (Figure 1). The model extends up to 40km in the vertical in 35 levels. Chemical boundary conditions to the Australia domain are provided by a global ACCESS-UKCA model run (Woodhouse et al., 2015).

Meteorological fields are provided by the Conformal Cubic Atmospheric Model (CCAM, r2796 (McGregor and Dix, 2008)), which is a global stretched grid dynamical model. CCAM predicts atmospheric dynamical conditions, including wind velocity, turbulence, temperature, radiation and the water vapour mixing ratio. The cloud coverage predicted by CCAM provides an attenuation factor which is applied to the photosynthetically active radiation (PAR) calculation.

The chemistry scheme is the extended Carbon Bond 5 mechanism (CB05) (Sarwar et al., 2011; Sarwar et al., 2008), consisting of 65 gas phase species, 19 aerosol species and 172 reactions. The organic species are lumped according to their carbon–carbon bonding type. CB05 combines individual monoterpenes into one lumped monoterpene species. Particulate species are processed in a two-bin sectional scheme with inorganic processing via ISORROPIA_II (Fountoukis and Nenes, 2007), and organic processing via the volatility basis set (Shrivastava et al., 2008). A 5 minute chemical timestep is used and all species are output on an hourly averaged basis.

Anthropogenic emissions come from the Sydney GMR inventory (NSW Department of Environment, Climate Change and Water (DECCW, 2007)) and includes 37 species. Anthropogenic sources include on- and off-road mobile, commercial, domestic and industrial point sources.

The C-CTM is set-up using two biogenic emission configurations; ABCGEM described in this paper, and MEGAN (Guenther et al., 2012) the set-up of which is described in Emmerson et al. (2016). The chemistry scheme and meteorological inputs are the same for both configurations, removing both as factors in possible model differences. Differences in the inputs required by each model are given below and in Table 1.

2.3.1 **ABCGEM model setu**p

The vegetation class used in ABCGEM is eucalypt forest, with the proviso that the canopy height and LAI are independent variables. ABCGEM requires an LAI dataset for the canopy to calculate the column biomass per unit ground area, $B_m$. These data are from Lu et al. (2003) and based on Advanced Very High Resolution Radiometer Normalised Difference Vegetation Index data between 1981 and 1994. Native Australian trees are evergreen therefore an annual average LAI is used, with a peak of 6.1 $m^2$ $m^{-2}$ (shown in the supplementary material). This yields a maximum fractional grid cell coverage of 0.95, occurring to the north west of the inner domain. The urban region of Sydney has a low tree LAI of between 1 - 2 $m^2$ $m^{-2}$. Note that the MUMBA inner domain is positioned further south than the other campaign domains, and the peak tree LAI in this grid is 4.6 $m^2$ $m^{-2}$. In ABCGEM, isoprene is treated as light and temperature dependent, whereas monoterpenes are treated as temperature dependent only, see Table 1. This monoterpene relationship is consistent with He et al's (2000) study of 15 eucalypts in Australia, where they found four of the strongest emitting species showed strong exponential temperature dependent relationships, three with an $r^2$ in excess of 0.9. While the range of PAR investigated was limited, He et al. (2000) found no relationship of eucalypt monoterpene emissions with PAR. There are significant differences between the light and temperature activity functions used in ABCGEM and MEGAN as part of the transformation of emission factors to emission estimates (supplementary material).

It is important to retain the original features of ABCGEM, including LAI, to provide an uncertainty estimate between two independently developed models on BVOC emissions in Australia. The total combined uncertainty of ABCGEM isoprene emissions at 95% confidence limits is approximately a factor of 2 (calculated in the supplementary material). However, to remove LAI as a cause of differences in the comparison, we also run ABCGEM replacing the LAI dataset with the MODIS dataset used with MEGAN (see the following section). This sensitivity test is referred to as 'AML'. Comparing MEGAN with AML ensures that the differences will only be due to each model's emission scheme. Comparing ABCGEM with AML shows how much of the emission uncertainties are due to choice of LAI dataset.

**2.3.2 MEGAN model setup**

MEGAN version 2.1 (Guenther et al., 2006; Guenther et al., 2012) is coupled to the C-CTM as an option for calculating 147 BVOC emission rates (Emmerson et al., 2016). The vegetation classes used in MEGAN are embedded within plant functional types and emission factor maps as described in Emmerson et al. (2016). Vegetation data comes from an International Global Biosphere Product (Belward et al., 1999) split into 16 plant functional types (PFTs) described in Emmerson et al (2016). Globally averaged emission factors are used to calculate the majority of MEGAN emissions, but emission factor maps are used for isoprene, myrcene, sabinene, limonene, 3-carene, ocimene, α-pinene, β-pinene, 2-methyl-3-buten-2-ol and NO at 1 km resolution. The MEGAN emission factor maps for Australia were produced by combining the mapped vegetation from Forests of Australia data (DAWR, 2003), with measurements of isoprene and monoterpene emission rates from Australian native plant species, as described in Emmerson et al. (2016). Regions where there is a large gradient in emission factors indicates a change in tree species or PFT. There are 41 monoterpene species in MEGAN, of which seven are mapped species listed above. All 41 are lumped together inside the C-CTM as a single monoterpene species according to the CB05 chemistry scheme. MEGAN uses monthly LAI data provided by MODIS MCD15A2 version 4. Details of the MEGAN emission equations in this CSIRO set-up are repeated in the supplementary material.

In MEGAN all species, including monoterpenes, have a light dependency (Guenther et al., 2012), which were set using global average behaviours. Measurements of α-pinene fluxes in the tropics do show a light dependence (Rinne et al., 2002), whereas emissions from boreal pine forests and some eucalypts are well described using a temperature dependent function only (Tarvainen et al., 2005; He et al., 2000). For the major monoterpene species, α-pinene, the light dependent function (LDF) in MEGAN is 0.6, where 1 represents complete light dependency (e.g. isoprene). For other monoterpenes in MEGAN the LDF

ranges between 0.2 – 0.8 (Guenther et al., 2012). This means that a proportion of the MEGAN monoterpene emissions shut off at night, whereas in ABCGEM they do not, and there will be differences in the emission processing during the day. To investigate these impacts, a sensitivity test will switch off the light dependence of all monoterpene species in MEGAN, referred to as "MEGAN-LDO". A discussion of the differences in the light and temperature activity functions between ABCGEM and
MEGAN is given in the supplementary material.

Guenther et al. (2012) estimate uncertainties in MEGAN isoprene emissions of a factor of 2, and for monoterpenes a factor of 3. They note that in regions with few observations such as Australia, these uncertainties could be higher.

## 3  Results and Discussion

### 3.1 Emission factors as a function of LAI

In Emmerson et al. (2016), we concluded that high emission factors controlled the over-estimation of isoprene in MEGAN. Figure 2 shows the role of projected LAI, using it to sort the mapped MEGAN isoprene and monoterpene emission factors in the 3km domain in February. ABCGEM uses constant emission factors described in section 2.2.1, which are converted to area units ($\mu$g m$^{-2}$ h$^{-1}$) using the $B_m$ weighted by LAI, in 1 m$^2$ m$^{-2}$ bins. Here LAI is weighted by the fractional area taken up by each bin. The percentage of land area covered by each LAI bin is also shown. We omit factors where the land area represents
less than 1% of the model domain. The equivalent plots for April (autumn) are shown in the supplementary material.

The ABCGEM emission factors are linearly dependent on LAI. The ABCGEM isoprene emission factors are generally lower than MEGAN, but within the MEGAN standard deviations. The positive standard deviations show MEGAN isoprene emission factors reaching 20 mg m$^{-2}$ h$^{-1}$, whereas the equivalent in ABCGEM would require an LAI above 7 m$^2$ m$^{-2}$. In MEGAN there is a distinct maximum at 3 - 4 m$^2$ m$^{-2}$ after which the emission factors decrease. Eucalypts are the major tree species around
Sydney occupying these 3 – 4 m$^2$ m$^{-2}$ regions of MODIS LAI, and are assigned the highest emission factors up to 24 mg m$^{-2}$ h$^{-1}$, causing the peak in Figure 2. The highest MODIS LAI is south of Sydney, and overlaps with regions of 'no data' in the Forests of Australia dataset surrounding a patch of temperate rainforest. These 'no data' regions are assigned low isoprene emission factors less than 3 mg m$^{-2}$ h$^{-1}$, as are urban areas. This mixture of high and low emission factors for the MODIS LAI range 4 - 6 m$^2$ m$^{-2}$ gives a reduced average emission factor causing the downturn. This is an illustration of the deficiencies in
vegetation mapping adversely affecting BVOC emissions modelling, similar to the findings of Arneth et al. (2011), Zhao et al. (2016), Huang et al. (2015), Otter et al. (2003), Warneke et al. (2010) and Langford et al. (2010).

The MEGAN monoterpene emission factors plotted are the sum of the mapped species (myrcene, sabinene, limonene, 3-carene, ocimene, $\alpha$-pinene and $\beta$-pinene) and represent most of the total monoterpene mass. The monoterpene emission factors for ABCGEM and MEGAN are similar below 3 m$^2$ m$^{-2}$ LAI, after which ABCGEM diverges, and is 39 % higher than MEGAN
at an LAI between 4 - 5 m$^2$ m$^{-2}$. However the influence of the highest ABCGEM emission factors is reduced as the percentage of grid cells occupied by LAI 4 – 5 m$^2$ m$^{-2}$ is 7 %. (Figure 2, right). There is the same downturn in MEGAN monoterpene emission factors at high LAI as for isoprene, for the same reasons given above. The standard deviations in MEGAN monoterpenes are much less than for isoprene. As the bulk of the land area is occupied by LAI less than 4 m$^2$ m$^{-2}$, the ABCGEM and MEGAN monoterpene emission factors are similar.

LAI is a key input factor to both models, but has more influence on BVOC emission factors in ABCGEM as the fractional areas covered by vegetation are controlled by the LAI. In MEGAN these fractional areas are controlled by the PFT maps. Broadleaf evergreen trees (Eucalypts) occupy up to 95% of the non-urban region of the Sydney model domain (Emmerson et

al., 2016). These emission factors are processed by emission activity functions incorporating radiation, temperature, LAI and PFT datasets, with both spatial and temporal differences, to calculate the emission fluxes (hereafter 'emissions').

## 3.2 Temporal differences in emissions

Domain average emissions for isoprene and monoterpenes are plotted as time series for the duration of each field campaign in Figure 3. For isoprene, there are days where ABCGEM and MEGAN give comparable results (~±20%) whereas there are other days when the isoprene emissions in MEGAN are more than double those of ABCGEM. This variation can be traced to the different activity functions in the two models as shown in the supplementary material. For temperatures below 305 K and PAR below 600 $\mu$mol m$^{-2}$ s$^{-1}$ the isoprene activity functions in the two models are comparable, whereas at higher temperatures and higher PAR the functions widely diverge; higher PAR favouring higher isoprene emissions in MEGAN and higher temperatures favouring higher isoprene emissions in ABCGEM. The impacts of these activity factors affects not only day to day variability in individual campaigns but also campaign to campaign differences.

In summer the daytime isoprene emissions from MEGAN are up to three times higher than ABCGEM or AML, whereas there was some overlap in their emission factors. This demonstrates the impacts of the lower radiation activity function in ABCGEM compared with MEGAN at summer noon PAR. The difference between MEGAN and ABCGEM is less in autumn for SPS2 when reduced temperatures and PAR cause substantial overlap in the SPS2 isoprene emissions. The MEGAN-LDO test has not affected the emissions of isoprene.

Isoprene and monoterpene emissions produced from the AML sensitivity run are 10% and 20% respectively different from ABCGEM and suggests that the choice (and age) of the LAI dataset is not critical to the BVOC emission estimates.

Whilst the monoterpene emission factors are similar between ABCGEM and MEGAN, the lower MEGAN monoterpenes are impacted by the light dependence of the MEGAN monoterpene activity function, see Table 1 and the Supplementary Material. Switching off the monoterpene light dependence in MEGAN increases the night time monoterpene emissions by 90 – 100% in MEGAN-LDO, making them comparable in magnitude to the ABCGEM and AML emissions. This is important in the model, as these night time emissions occur when the boundary layer is shallow, and the chemical removal processes are much slower. MEGAN-LDO shows a minor increase in the daytime peak monoterpene emissions compared to MEGAN. The emissions of monoterpenes in MEGAN or MEGAN-LDO during the day do not reach the same magnitudes as those from ABCGEM, as the MEGAN emission flux is not due to the temperature activity function alone. Activity functions for LAI and the leaf age also play a role. However the chemical removal processes for monoterpenes during the day are much stronger, so it is expected that the differences in daytime emission fluxes between MEGAN and ABCGEM are less discernible in daytime measurements at the field campaign sites.

## 3.3 Spatial distribution of emissions

The spatial distribution in the emissions are now examined using the SPS1 campaign as an example. Figure 4 shows maps of the grid cell average emissions for ABCGEM, AML and MEGAN, followed by the differences between them. The difference plots subtract the ABCGEM or AML emissions from MEGAN, where red shows positive differences (MEGAN higher) and blue shows negative differences (MEGAN lower). The difference between MEGAN and MEGAN-LDO for monoterpenes is only up to 95 g km$^{-2}$ h$^{-1}$ (not shown) as it is mainly the lower emissions at night time that have increased as shown in Figure 3. Equivalent maps for SPS2 are shown in the supplementary material to demonstrate the seasonal differences.

The SPS1 peak isoprene emission of 6473 g km$^{-2}$ h$^{-1}$ for MEGAN occurs to the north west of Westmead in the Blue Mountain ranges, matching with the location of the highest emission factors. The peak ABCGEM isoprene emission of 2441 g km$^{-2}$ h$^{-1}$

occurs to the north east of Westmead (near Wyong), at the location of the highest projected LAI. The AML peak isoprene emission occurs in the same location as ABCGEM due to high LAI here, but is slightly lower at 2391 g km$^{-2}$ h$^{-1}$. Where MEGAN shows inland patches with no emissions, these are due to zero emission factors at these locations, e.g. Lake Burragorang, west of Bringelly. ABCGEM relies entirely on the LAI distribution to place the emissions, and neither the

5 ABCGEM nor MODIS LAI distribution recognise these lake features. This again is an illustration of the deficiencies in vegetation mapping adversely affecting BVOC emissions modelling.

In the isoprene difference plots, MEGAN predicts 1000 – 4000 g km$^{-2}$ h$^{-1}$ more isoprene to the west and north of Sydney than ABCGEM/AML, an increase of 40 – 200%. However MEGAN predicts 100 – 1000 g km$^{-2}$ h$^{-1}$ less isoprene than ABCGEM/AML in the urban regions where the field campaigns took place, contrary to the domain averages (at Westmead

MEGAN is 15% lower, at Randwick, 46% lower). In this urban zone, MEGAN has a low fraction of plant coverage (30%) and an isoprene emission factor less than 3 mg m$^{-2}$ h$^{-1}$ associated with urban deciduous trees. In ABCGEM (and AML) the urban fraction of plant coverage and emission factors are dependent on the projected LAI which is 1 - 2 m$^2$ m$^{-2}$ here. Thus ABCGEM vegetation covers a larger area of the urban grid cells (39 - 63%), and the corresponding emission factor, being for eucalypts, is also larger (2.8 - 5.7 mg m$^{-2}$ h$^{-1}$, or up to 47%) than MEGAN. These spatial patterns reiterate that a key difference

between the two isoprene emission models is the input vegetation type and coverage.

The peak ABCGEM monoterpene emission of 1701 g km$^{-2}$ h$^{-1}$ also occurs in the north east of the domain (near Wyong), and is more than three times the peak monoterpene emission at the same location in MEGAN. ABCGEM and AML predict between 0 - 300 g km$^{-2}$ h$^{-1}$ more than MEGAN over most of the domain. The only location where MEGAN predicts higher monoterpene emissions than ABCGEM, occurs about 30 km southwest of Sydney (shown in red, Figure 4d). MEGAN predicts 0 – 300 g

20 km$^{-2}$ h$^{-1}$ more monoterpenes than ABCGEM, but this difference is not observed between MEGAN and AML and must result from a difference in the LAI dataset. At this location, the ABCGEM LAI is 0.6 m$^2$ m$^{-2}$ and is considered to be 'urban'. The MODIS LAI is 3 m$^2$ m$^{-2}$ and corresponds with a region of "Eucalypt medium woodland" in the Forests of Australia inventory (on which the MEGAN emission factors are based). This same feature is present for isoprene, though is less visible in Figure 4c because differences elsewhere in the domain are also large. These differences are due to the spatial distribution of the

25 different LAI datasets used by ABCGEM and MEGAN.

Geometric mean emissions are calculated for each of the models and presented in table 2. The MEGAN isoprene emissions are a factor of 1.7 larger (range 1.4 – 2.1) than ABCGEM across the five field campaigns, with the higher values occurring in summer. As the AML isoprene emissions are a factor of 0.9 times lower than ABCGEM, MEGAN is a factor of 1.9 higher (range 1.5 - 2.5) than AML. For monoterpenes, the ABCGEM emissions are larger than MEGAN by a factor of 2.1 (range 2.0

30 – 2.4), with the larger values tending towards autumn. AML monoterpenes are a factor of 1.7 higher (range 1.5 -1.9) than MEGAN, and a factor of 0.8 lower than ABCGEM.

Table 2 presents the ratio of isoprene to monoterpene carbon for these geometric mean emissions. Emmerson et al. (2016) found ratios close to 1 for observed levels in the Sydney basin. This is in contrast to a ratio of 0.18 found in boreal forests dominated by monoterpenes (Spirig et al., 2004), and to a ratio of 26.4 in deciduous Michigan forests dominated by isoprene

(Kanawade et al., 2011). SOA formation is inhibited in regions where isoprene dominates, however it is not known what impact a carbon ratio of 1 will have. The carbon ratio is most likely controlled by metabolic processes within the plants and as such is a valid test of the models. The biochemistry behind this competition is explained in Harrison et al. (2013) who present emission capacities from species worldwide emitting both isoprene and monoterpene. Two thirds of the 80 cases have ratios greater than 1. Monoterpene emissions are favoured in nitrogen poor conditions (Fernández-Martínez et al., 2017) in species

with a long leaf lifespan (Harrison et al. 2013), conditions matching Australia.

The average carbon ratio for ABCGEM is 1.3 (range 0.8 – 1.5), AML is also 1.3 (range 0.9 – 1.6), whilst the MEGAN ratio is higher at 4.4 (range 2.7 – 5.3). As it is mainly the lowest monoterpene emission fluxes that have increased in the MEGAN-LDO test, the geometric mean emissions have not increased much from the MEGAN test (6% - 50% as season tends towards autumn), resulting in minor improvements to the MEGAN-LDO average carbon ratio for emissions (4.0, range 1.8 – 5.3).

Whilst these ranges demonstrate the substantial uncertainties in the estimated emissions, the ABCGEM and AML ratios are more in line with Australian observed isoprene to monoterpene carbon ratios.

**3.4 Predicted versus observed atmospheric volume mixing ratios**

Isoprene and monoterpenes from the time periods and locations of each field campaign have been extracted from the models to compare with the PTR-MS observations. The transport and chemical schemes are the same in each model therefore for any

particular campaign, the bulk of the differences between the ABCGEM and MEGAN models should directly scale to the differences in emissions between the models. Campaign average diurnal cycles are shown in Figure 5, with the percentage of points within a factor of 2 of the observations.

ABCGEM predicts isoprene and monoterpene levels closer to those observed compared to MEGAN. In all of the 10 cases ABCGEM predicts an equal or higher number of points within a factor of two of the observations than MEGAN. AML

generally predicts higher isoprene and monoterpene levels for all campaigns than ABCGEM because the campaign sites are within the urban zone where the MODIS LAI is higher than the ABCGEM LAI.

Usually isoprene peaks with solar noon, but the modelled and observed isoprene at Randwick peaks at 9 am, decreasing afterwards (Figure 5). All models show isoprene increasing after 7pm which suggests the phenomena is not a function of the emission model, but of the meteorology. A stable nocturnal boundary layer develops post 7 pm. The isoprene decrease after

9am at Randwick is due to a change in wind direction, bringing marine air with low BVOCs to the Randwick site. Randwick is close to the coast, therefore local isoprene emissions do not build up with easterly winds. Peak monoterpene levels occur at night, so are not affected by the daytime onshore breezes. The MUMBA campaign site is also coastal. The wind direction at MUMBA switches from south west to north east later in the day, travelling over land regions and allowing isoprene to be present during the day. Wind roses have been plotted for each of the campaigns and are shown in the supplementary material,

along with a detailed hourly analysis for MUMBA and Randwick. These support the above analysis.

The daytime ABCGEM and AML isoprene levels estimated for SPS2 are greater than those predicted by MEGAN. Whilst the domain average isoprene emissions in ABCGEM are lower than MEGAN for SPS2, the ABCGEM isoprene emissions in urban areas are higher in ABCGEM than MEGAN. This arises because MEGAN uses a lower emission factor (deciduous trees) in urban areas than ABCGEM which only has eucalypts, in combination with ABCGEM having a larger vegetation

cover of grid cells in urban areas compared with MEGAN, as discussed previously.

Changes to the oxidants as a result of the additional monoterpenes in the MEGAN-LDO test has impacted on the isoprene at the campaign sites, in general reducing MEGAN daytime isoprene by 4% and night time isoprene by 15%. MEGAN-LDO has also improved the percentage of points within a factor of 2 of the observations for isoprene. This is not the case for isoprene at MUMBA which has increased during the daytime by 55% and at night by 18%, reducing the percentage within a factor of

2 of the observations to 4%. This is because the monoterpene levels in the MEGAN-LDO test have increased by 163% at night and 65% during the day over the very hot January 2013 of the MUMBA campaign, more than for any other field campaign, impacting the oxidant chemistry. Peak modelled OH for MUMBA has decreased by 0.1 ppt (~1700%) and $HO_2$ by 1.5 ppt (~350%).

At all sites except Bringelly, ABCGEM represents the shape and magnitude of the observed monoterpene diurnal cycles well, whilst MEGAN under-predicts. However the night-time monoterpene emissions have increased in the MEGAN-LDO test compared with MEGAN and we see increased night-time monoterpenes at all the campaign sites, on average by 61%. This is consistent with the light independent activity function leading to higher monoterpene emissions and volume mixing ratios and is more in line with these observations. The daytime monoterpene levels are similar in ABCGEM and MEGAN-LDO

despite the large difference in daytime emissions due to the strong chemical processing. The daytime increase in monoterpenes between the MEGAN and the MEGAN-LDO test is 25%. Monoterpene storage pools in Australian native vegetation may behave differently to the average global conditions represented in MEGAN, and in-situ observations in Australia are necessary to determine the process correctly. Average monoterpene emissions for SPS1 are of a similar magnitude at Bringelly, yet the

observed monoterpenes at Bringelly are half those observed for SPS1, resulting in a large over-prediction at Bringelly by all models. Light dependence is not the only issue at Bringelly, where the model is more influenced by stronger winds from the west and north than the observations, resulting in higher modelled BVOCs than observed. Further wind rose analysis is given in the supplementary material.

Table 3 gives the campaign average temperatures and atmospheric volume mixing ratios from the models and the observations.

We also include the observed isoprene to monoterpene carbon ratios, which were presented in Emmerson et al. (2016) for SPS1, SPS2 and MUMBA. The observed carbon ratios for Bringelly (1.5) and Randwick (1.0) datasets roughly conform to the unity phenomena in SE Australia with all the measurements giving an average of 1.2 (range 0.9 – 1.5). In the models the average carbon ratio across all campaigns is 1.2 (range 0.7 – 1.7) for ABCGEM, 1.0 (range 0.6 – 1.4) for AML, 4.1 (range 1.7 – 7.3) for MEGAN and 2.6 (range 0.8 – 4.7) for MEGAN-LDO, similar to the emission results. The reductions in carbon ratio

due to the MEGAN-LDO test show that increasing the night time monoterpene level by switching off the light dependence improves this relationship. The carbon ratios for ABCGEM and AML are more within the observed range, suggesting the balance between isoprene and monoterpene emissions are about right. Improvements to MEGAN should concentrate on this balance.

Figure 6 shows a quantile-quantile (q-q) plot, where all modelled and observed data from all five field campaigns are paired

in time and ranked from low to high volume mixing ratios, forming one line per sensitivity run. Logarithmic axes are chosen as the region below 1 ppb represents 93% of the observed data points (for observed isoprene 24% are between 0.01 - 0.1 ppb, 69% are between 0.1 - 1 ppb with only 7% above 1 ppb). For ease of comparison, a 1:1 line is plotted. All models predict isoprene levels that are too low at observations < 0.3 ppb, after which all models over-predict.

Normalised mean biases (NMB) have been calculated comparing each emission model to the ranked observations (equation

1), where P are the predicted levels from the model and O are the observed levels. An NMB closer to zero is regarded as the better comparison.

$$NMB = \frac{\sum(P-O)}{\sum O} \tag{1}$$

The MEGAN-LDO test has improved the isoprene bias for the lowest 50% of the isoprene data points from 0.01 to 0.28 ppb of observed isoprene, whilst AML is most biased. ABCGEM is less biased for the upper 50% of data points from 0.28 – 7.1

35  ppb of observed isoprene, and MEGAN is most biased. With the inputs used in this study, the overall NMB for isoprene is 0.45 for ABCGEM, compared to 0.67 for AML, 1.39 for MEGAN and 1.58 for MEGAN-LDO.

The monoterpene q-q plot has been clipped to observations > 0.04 ppb, which was the instrument limit of detection at Bringelly and Randwick. ABCGEM is less biased for the first 90% of data points from 0.04 – 2.7 ppb of observed monoterpene; MEGAN

least biased for the last 10%. AML is more biased than ABCGEM and tends to over-predict at all observed levels, whilst MEGAN mainly under-predicts. The MEGAN-LDO test has improved the bias below 0.3 ppb of observed monoterpenes, but degraded the bias above 1 ppb, where previously MEGAN was least biased. Overall, the monoterpene NMB for ABCGEM is 0.33, 0.56 for AML, -0.28 for MEGAN and 0.24 for MEGAN-LDO. NMB calculations for each field campaign considered individually are shown in table 3.

One goal in this work is to calculate a total uncertainty in BVOC emissions for the Sydney GMR. Two approaches are used in this paper. In section 2.3.1 a bottom up uncertainty assessment for ABCGEM (presented in the supplementary material) was discussed. Here a top-down assessment is made using the calculated normalised mean biases between the models and observations in table 3. These provide the scatter from model to model and campaign to campaign as a measure of uncertainty. The 95% confidence limits from the NMBs in table 3 are equivalent to uncertainties of factors of ~2 for isoprene and ~3 for monoterpenes. This is consistent with the estimate of a factor of 2 from the bottom up estimate that omits uncertainty due to knowledge missing from the models, and also consistent with the factors of 4 difference in the modelled carbon rations between ABCGEM and MEGAN.

## 4    Conclusions

The purpose of this work was to uncover reasons for the discrepancies produced by MEGAN in modelling BVOCs in the south east Australian region identified by Emmerson et al. (2016). This is a largely unstudied region with very few measurements of BVOC emissions. By comparing the locally developed ABCGEM and the well-established MEGAN model, both in terms of estimated emissions and also via simulated and observed atmospheric volume mixing ratios of isoprene and monoterpenes, we use local knowledge to suggest improvements for the application of MEGAN in Australia. Both models are run within the C-CTM, for five field studies within the Sydney GMR, in New South Wales, Australia. Both models use the same meteorology and chemistry scheme from the C-CTM but each have independent inputs for LAI and BVOC emission factors. We examined the differences in the LAI input by running ABCGEM with both LAI inputs and found small differences of 10% and 20% in isoprene and monoterpene emissions, respectively.

Emmerson et al. (2016) concluded that the MEGAN emission factors may not be appropriate for south east Australia. However similar emission factors used in ABCGEM suggest this may not be the case and it is the processing of these emission factors that should be investigated. The isoprene emission factors used in MEGAN are in a similar range to the LAI-weighted ABCGEM emission factors, but the MEGAN standard deviations extend much higher than ABCGEM. The eucalypt trees surrounding Sydney have a projected LAI in the 3 - 4 $m^2\,m^{-2}$ region, where MEGAN isoprene emission factors are about 50% higher than ABCGEM. For monoterpenes, the ABCGEM emission factors increase linearly, whilst the MEGAN emission factors peak at 3 $m^2\,m^{-2}$ thereafter decreasing due to averaging of high and low emission factors at high LAI. As the bulk of the LAI in the 3km domain is less than 4 $m^2\,m^{-2}$, the ABCGEM monoterpene emission factors are similar to MEGAN.

There are differences in the temperature and radiation activity functions between ABCGEM and MEGAN, causing MEGAN to produce more isoprene and less monoterpenes than ABCGEM on an Australian summer's day, if all other inputs are equal. Using the geometric mean emissions, the MEGAN isoprene emissions across the five field campaigns are a factor of 1.7 larger (range 1.4 – 2.1) than ABCGEM and a factor 1.9 larger (range 1.5 - 2.5) than AML.

The monoterpene emission factors are similar between the models, but the resulting emission fluxes are very different because MEGAN has a light dependence whereas ABCGEM does not. MEGAN monoterpene emissions are lower than ABCGEM by a factor of 2.1 (range 2.0 – 2.4), and lower than AML by a factor of 1.7 (range 1.5 -1.8). We also tested the impacts of switching

off the light dependence of monoterpene species in MEGAN, as motivated by measurements by He et al (2000) on Australian eucalypts. During summer, the night time monoterpene emissions are increased by 90 – 100 % with the light dependence disabled, compared with the standard MEGAN run.

The distribution of ABCGEM, AML and MEGAN emissions are spatially different, with ABCGEM and AML predicting peak
isoprene to the north east of Sydney, and MEGAN predicting peak isoprene to the north west of Sydney. In ABCGEM and AML the emission distributions are dependent on the LAI dataset, whereas in MEGAN the impact of LAI is less dominant than the emission factor maps. ABCGEM and AML predict more isoprene in urban regions than MEGAN, which is the influence of the switch from eucalypt to deciduous trees in urban areas in MEGAN. ABCGEM remains eucalypt covered and has a greater fractional vegetation coverage in urban areas than MEGAN.

The volume mixing ratios of isoprene and monoterpenes from the model runs were compared to PTR-MS observations made at each field campaign site. As the transport and chemical processing were the same in each model, the bulk of the differences in the model were due to the differences in calculated emissions. For four of the five campaigns the ABCGEM model predicts lower isoprene and higher monoterpenes than the MEGAN model. ABCGEM had a higher number of modelled points within a factor of 2 of the observations than MEGAN or AML for both isoprene and monoterpene comparisons. MEGAN tends to
under-predict levels of Australian monoterpenes by a factor of 3, which is improved by switching off the light dependence of monoterpene species. Monoterpenes from Australian vegetation may not be as light dependent as vegetation globally, and this can only be ascertained though in-situ measurements.

In south east Australia we are starting to see a trend of unity for campaign average observed ratios of isoprene to monoterpene carbon, not observed in other parts of the world. In this study we present two additional observed datasets conforming to this
phenomena. The ABCGEM model predicts isoprene and monoterpene levels producing an average carbon ratio of 1.2, and 1.0 for AML. MEGAN over-predicts isoprene and under-predicts monoterpenes to the extent that the average carbon ratio is 4.1, but by removing the light dependence of the monoterpene emission activity function increasing the night time monoterpenes the carbon ratio improves to 2.6. In ABCGEM this suggests the balance between isoprene and monoterpene emissions are about right, but there is still work to be done on the magnitudes of the MEGAN emissions.

We calculate a total uncertainty for Australian BVOC emissions of a factor of 2 for isoprene and a factor of 3 for monoterpenes, based on a combination of modelling and observations. This provides a guide to the uncertainty that might be expected in applying an emission model to a region where the BVOC emissions have not been observed or modelled previously.

These comparisons are undertaken to strengthen understanding and to identify ways to reduce uncertainty in emissions of isoprene and monoterpenes in Australia. We have highlighted the roles of the spatial and temporal distributions of LAI and
the correct mapping of plant species or plant functional types in this modelling. One of the main contributions of this work is the examination of the role of light dependence in monoterpene emissions, which have helped improve the MEGAN comparison with observations. Targeted measurements on in-situ Australian vegetation, particularly of the light dependence issue for both isoprene and monoterpenes are critical to improving MEGAN for one of the world's major BVOC emitting regions.

**Data Provision**

Observed PTR-MS data is available for SPS1 (Keywood et al., 2016a)(http://doi.org/10.4225/08/57903B83D6A5D), SPS2 (Keywood et al., 2016b) (http://doi.org/10.4225/08/5791B5528BD63) and MUMBA (Guérette et al., 2017)

(https://doi.pangaea.de/10.1594/PANGAEA.871982). The Bringelly and Randwick PTR-MS data are available from the author.

The MODIS LAI data product was retrieved from MCD15A2 version 4 from the online Data Pool, courtesy of the NASA Land Processes Distributed Active Archive Center (LP DAAC), USGS/Earth Resources Observation and Science (EROS)

Center, Sioux Falls, South Dakota, https://lpdaac.usgs.gov/data_access/data_pool.

## Acknowledgements

This work was funded by the Environmental Research Program of the Environment Trust of NSW through the "Atmospheric Particles in Sydney: model observation verification study", number 2014/RD/0029. KME acknowledges funding from the NSW Office of Environment and Heritage and the Clean Air and Urban Landscapes Hub, which is a project of the Department

of the Environment's National Environmental Science Program. The Bringelly and Randwick observations were made as part of the Clean Air Research Program, Department of the Environment, Water, Heritage and the Arts, Commonwealth of Australia. Thanks to Drs Ying-Ping Wang and Richard Smart for helpful discussions.

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

**Table 1 Input datasets and characteristics of ABCGEM and MEGAN modelling**

| | ABCGEM (this work) | MEGAN (Emmerson et al., 2016) |
|---|---|---|
| Meteorology, including temperature and PAR | CCAM | CCAM |
| Chemistry scheme | Carbon Bond 5 | Carbon Bond 5 |
| Anthropogenic emissions | GMR inventory (DECCW, 2007) | GMR inventory (DECCW, 2007) |
| LAI | Monthly grids (Lu et al., 2003) | Monthly MODIS files, for current and previous monthly LAI. |
| Plant Functional Type, PFT | 2 classes: trees and grass | 16 PFTs from IGBP dataset (Belward et al., 1999) |
| Emission factors, $EF_S$ | Mapped by weighting standard emission rates of 25 $\mu$g-C g$^{-1}$ h$^{-1}$ for isoprene and 2.5 $\mu$g-C g$^{-1}$ h$^{-1}$ for monoterpenes by column biomass | Mapped emission factors for 10 species, including isoprene and 7 monoterpene species; fixed values dependent on PFTs for the other 137 species |
| Activity Functions | Isoprene: light and temperature Monoterpenes: temperature only | All species: light, temperature, LAI and leaf age. |
| No. of layers in canopy model | 10 (8 above trunk) | 5 |
| Considers energy balance? | No | Yes |

**Table 2 Geometric mean emission fluxes, g km⁻² h⁻¹ for isoprene and monoterpenes across the five field campaigns. The difference between the geometric means is also given as a factor, using MEGAN/ABCGEM for isoprene and ABCGEM/MEGAN for monoterpenes. Result for AML and MEGAN-LDO given in brackets.**

| | Isoprene | | | Monoterpenes | | | Ratio isoprene to monoterpene carbon | |
| | ABCGEM (AML) | MEGAN | Difference M/A | ABCGEM (AML) | MEGAN (M-LDO) | Difference A/M | ABCGEM (AML) | MEGAN (M-LDO) |
|---|---|---|---|---|---|---|---|---|
| MUMBA | 1489 (1238) | 3123 | 2.1 (2.5) | 598 (435) | 297 (294) | 2.0 (1.5) | 1.2 (1.4) | 5.3 (5.3) |
| Bringelly | 1738 (1487) | 2849 | 1.6 (1.9) | 574 (474) | 283 (300) | 2.0 (1.7) | 1.5 (1.6) | 5.0 (4.7) |
| SPS1 | 1562 (1467) | 2767 | 1.8 (1.9) | 608 (534) | 295 (319) | 2.1 (1.8) | 1.3 (1.4) | 4.7 (4.3) |
| Randwick | 1385 (1159) | 2039 | 1.5 (1.8) | 486 (410) | 228 (278) | 2.1 (1.8) | 1.4 (1.4) | 4.5 (3.7) |
| SPS2 | 372 (339) | 516 | 1.4 (1.5) | 229 (180) | 96 (144) | 2.4 (1.9) | 0.8 (0.9) | 2.7 (1.8) |

**Table 3 Comparison of the observed atmospheric volume mixing ratios (vmr) of isoprene and monoterpenes to model estimates from ABCGEM and MEGAN. NMB= normalised mean bias. M-LDO = MEGAN-LDO. Observed and modelled average temperature (and range) shown.**

| | | Temperature | Isoprene | | Monoterpenes | | Ratio isoprene to monoterpene carbon |
|---|---|---|---|---|---|---|---|
| | | Average (range), K | Average vmr, ppb | NMB | Average vmr, ppb | NMB | |
| MUMBA | Observed | 295.1 (287.6 – 317.4) | 0.28 | | 0.12 | | 1.2 |
| | ABCGEM | 296.1 (287.0 – 315.6) | 0.34 | 0.18 | 0.14 | 0.12 | 1.2 |
| | AML | | 0.28 | 0.01 | 0.17 | 0.37 | 0.8 |
| | MEGAN | | 0.88 | 1.96 | 0.06 | -0.50 | 7.3 |
| | M-LDO | | 1.22 | 3.10 | 0.13 | 0.13 | 4.7 |
| Bringelly | Observed | 295.9 (284.1 – 308.9) | 0.48 | | 0.16 | | 1.5 |
| | ABCGEM | 296.6 (286.1 – 310.9) | 0.83 | 0.69 | 0.62 | 2.80 | 0.7 |
| | AML | | 0.97 | 1.18 | 0.75 | 3.10 | 0.6 |
| | MEGAN | | 1.47 | 1.55 | 0.43 | 1.32 | 1.7 |
| | M-LDO | | 1.00 | 1.28 | 0.67 | 2.7 | 0.8 |
| SPS1 | Observed | 295.6 ( 286.4 – 310.1) | 0.76 | | 0.44 | | 0.9 |
| | ABCGEM | 298.0 (289.6 – 315.9) | 1.00 | 0.37 | 0.36 | -0.17 | 1.4 |
| | AML | | 1.23 | 0.61 | 0.45 | 0.00 | 1.4 |
| | MEGAN | | 1.35 | 0.89 | 0.21 | -0.53 | 3.2 |
| | M-LDO | | 1.35 | 0.81 | 0.31 | -0.29 | 2.2 |
| Randwick | Observed | 294.0 (285.8 – 304.5) | 0.28 | | 0.14 | | 1.0 |
| | ABCGEM | 296.5 (291.8 – 308.0) | 0.37 | -0.22 | 0.11 | -0.50 | 1.7 |
| | AML | | 0.38 | -0.26 | 0.13 | -0.52 | 1.5 |
| | MEGAN | | 1.11 | 1.11 | 0.09 | -0.61 | 6.2 |
| | M-LDO | | 0.96 | 0.75 | 0.13 | -0.50 | 4.4 |
| SPS2 | Observed | 289.0 (277.1 – 300.6) | 0.54* | | 0.46 | | N/A* |
| | ABCGEM | 290.7 (281.8 – 301.4) | 0.72 | 1.02 | 0.37 | -0.16 | 1.0 |
| | AML | | 0.85 | 1.61 | 0.48 | 0.09 | 0.9 |
| | MEGAN | | 0.70 | 0.75 | 0.17 | -0.61 | 2.1 |
| | M-LDO | | 0.69 | 0.78 | 0.32 | -0.28 | 1.1 |

* SPS2 average observed volume mixing ratio of isoprene is different from Emmerson et al. (2016) because evening/night data has been removed due to wood smoke contamination.

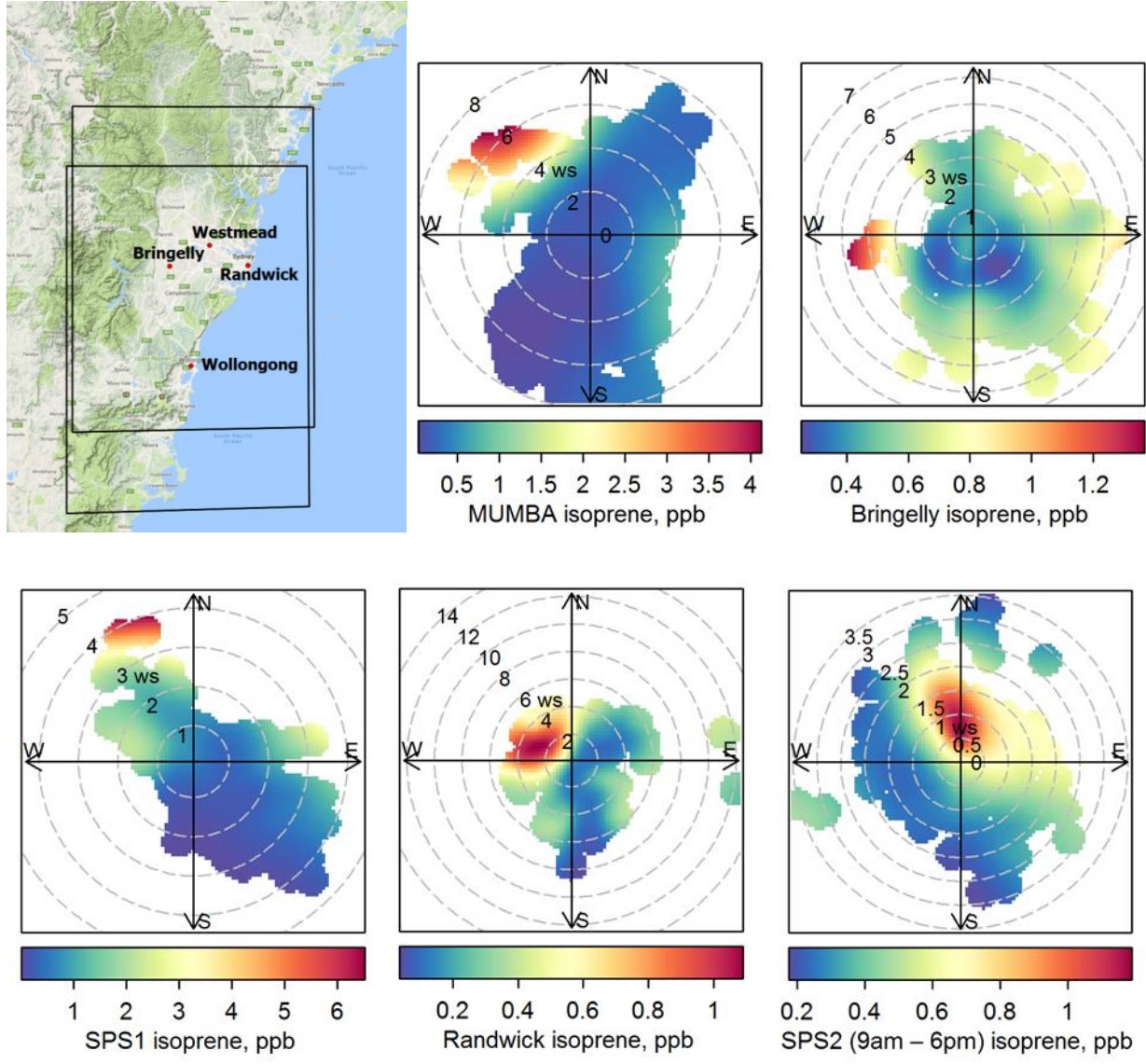

**Figure 1 Physical map of the Sydney Greater Metropolitan Region, and bivariate polar plots of isoprene observations from all field campaigns, arranged by time of year (summer to autumn). Map shows the position of the field campaign sites in relation to the surrounding forested regions, and the extent of the 3km inner domains. Map produced by QGIS using Google physical layer. Openair used to make bivariate polar plots (Carslaw and Ropkins, 2012).**

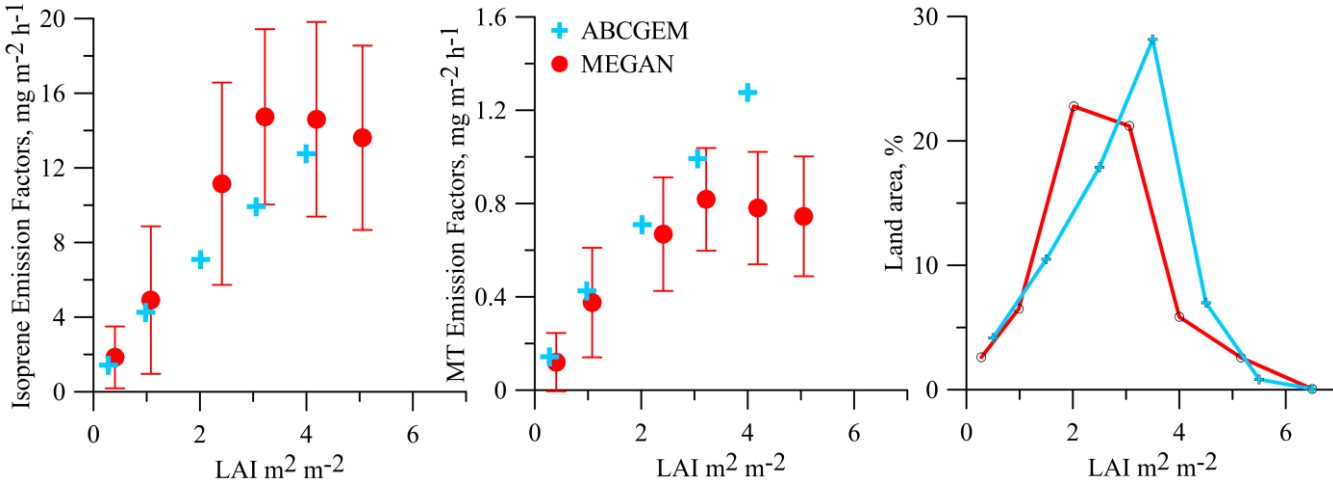

**Figure 2 Scatter plot of the canopy isoprene (left) and monoterpene (MT, middle) emission factors across the Sydney domain with LAI for ABCGEM and MEGAN during February. Note y-axes are not the same. (Right) percentage of land area within each LAI bin in February. Error bars represent ± 1 standard deviation.**

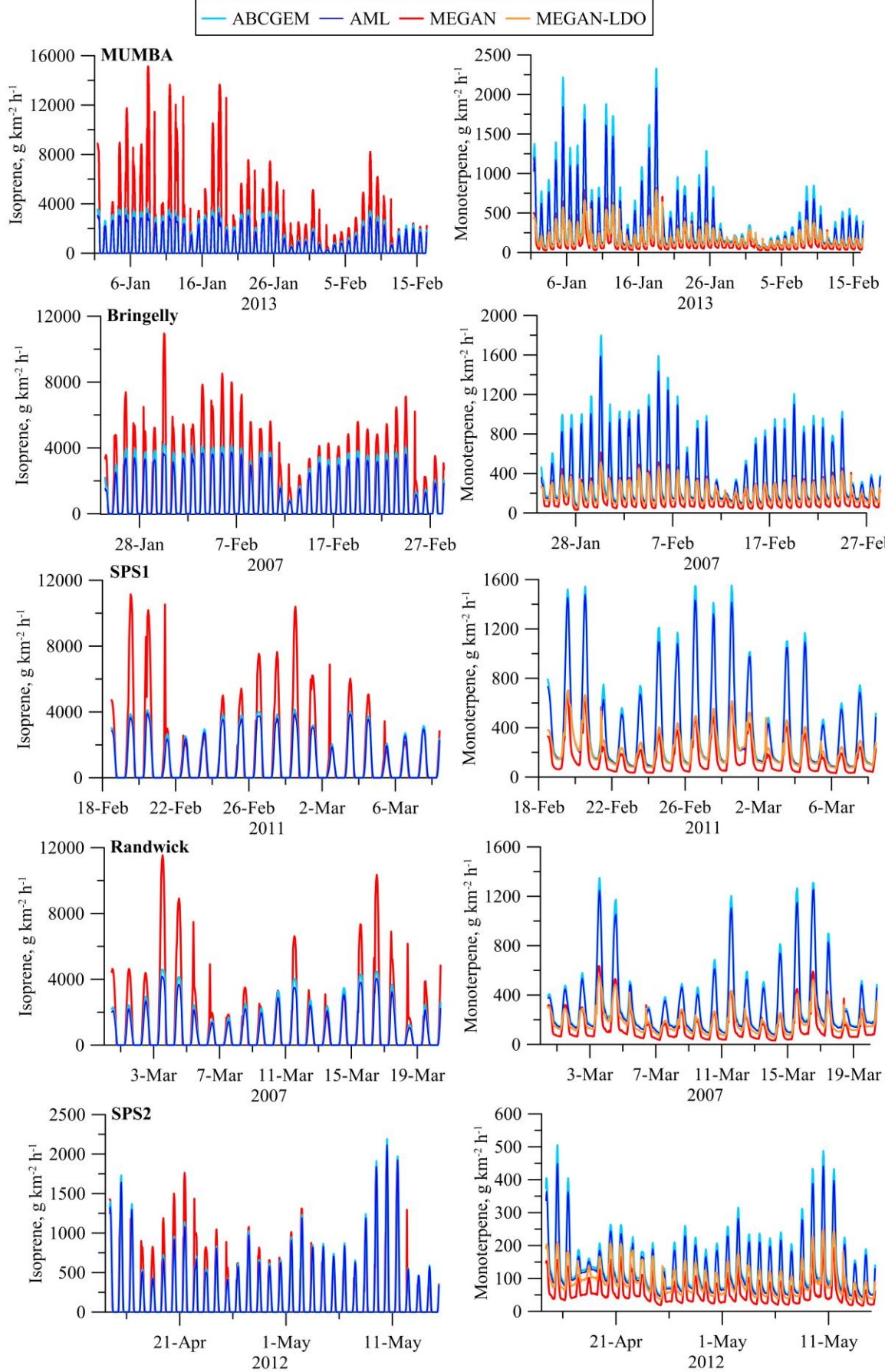

**Figure 3 Time series in domain average emission fluxes for isoprene (left) and monoterpene (right).**

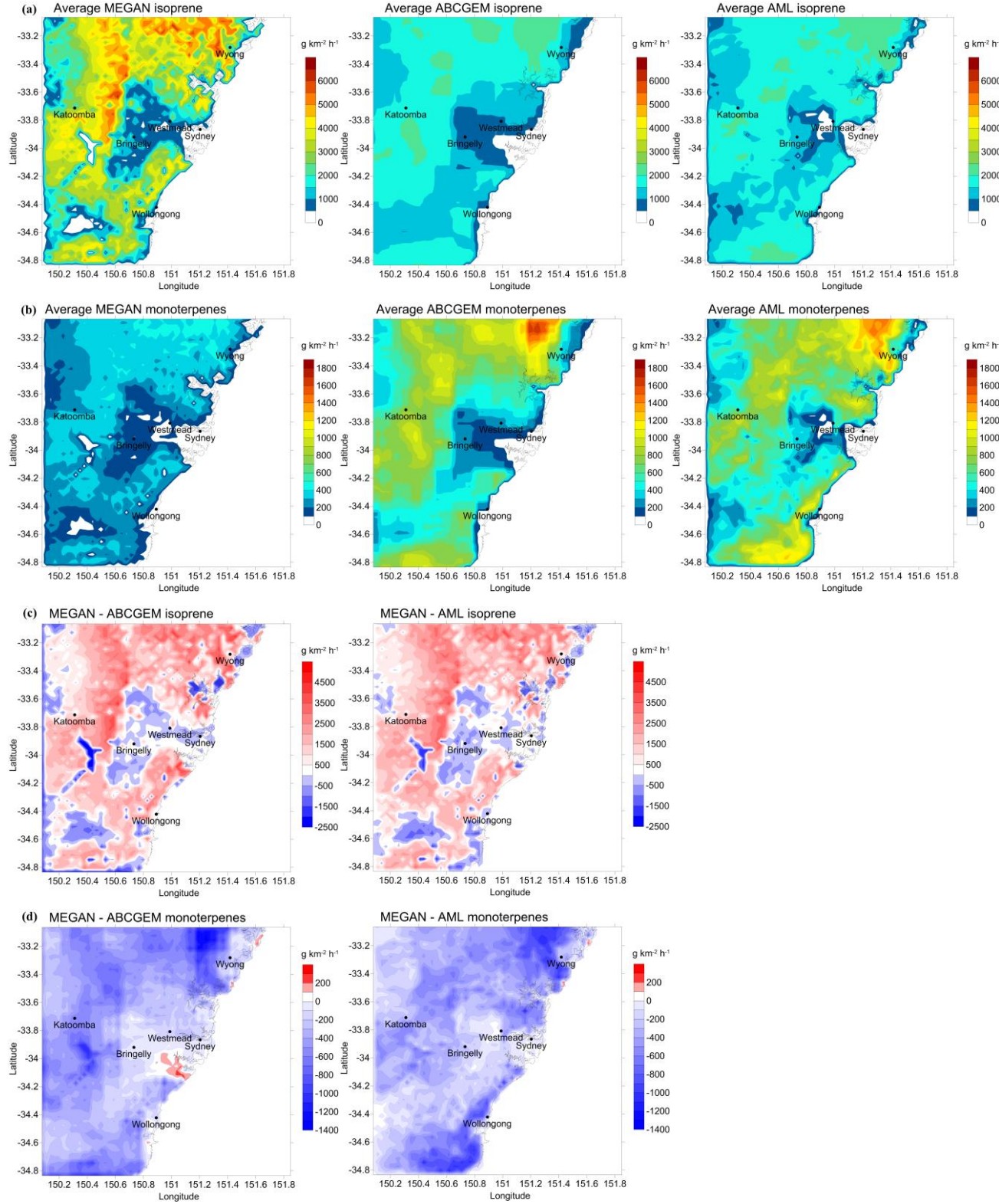

5 **Figure 4 Spatial distributions of grid cell average emission fluxes for (a) isoprene (b) monoterpenes, and the differences between MEGAN with ABCGEM or AML emission fluxes for (c) isoprene and (d) monoterpenes for the SPS1 campaign. Note: scales are unlike for isoprene and monoterpenes.**

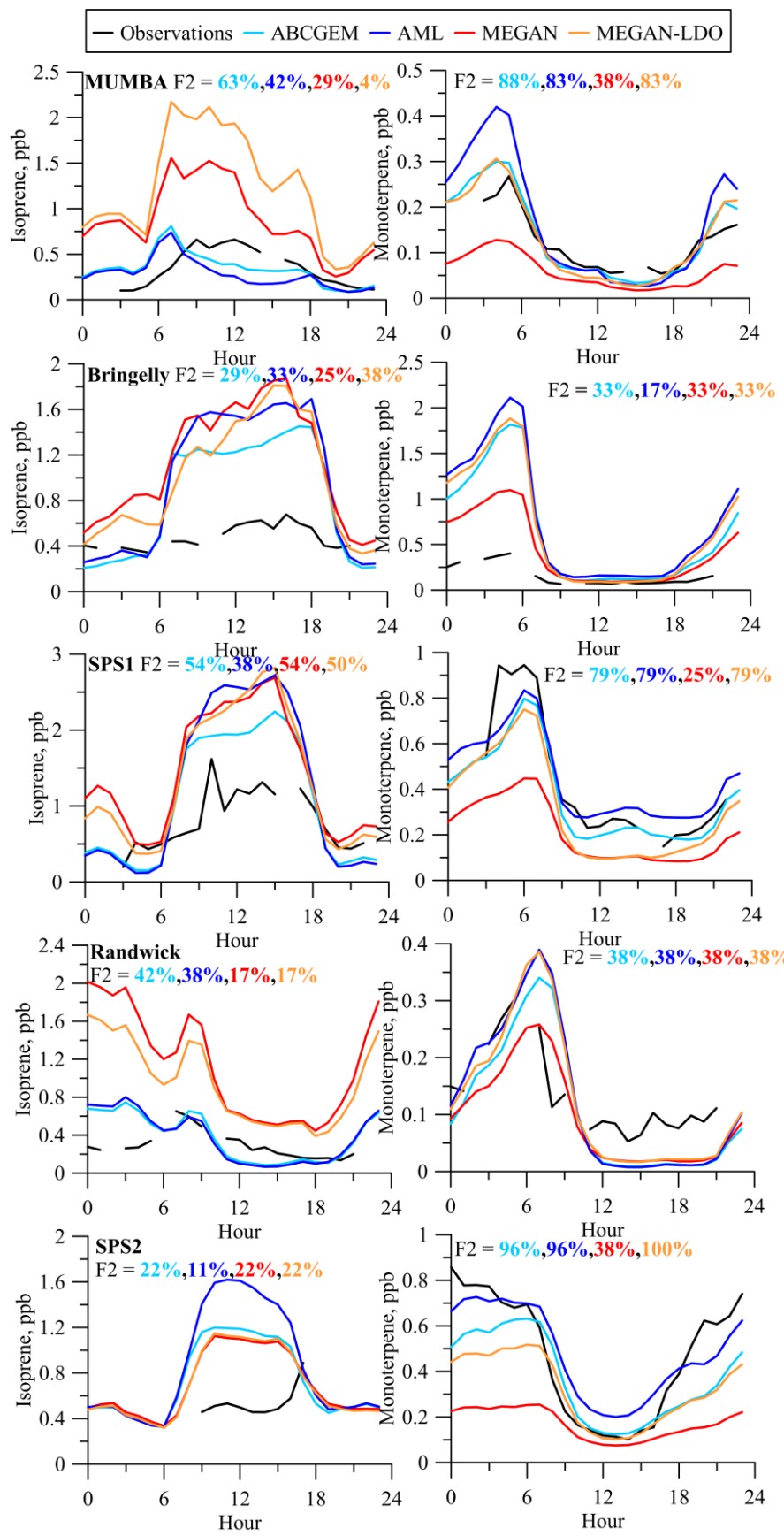

**Figure 5 Diurnal time series of modelled and observed isoprene (left) and monoterpenes (right) at MUMBA, Bringelly, SPS1, Randwick and SPS2. F2 is the percentage of points within a factor of 2 of the observations.**

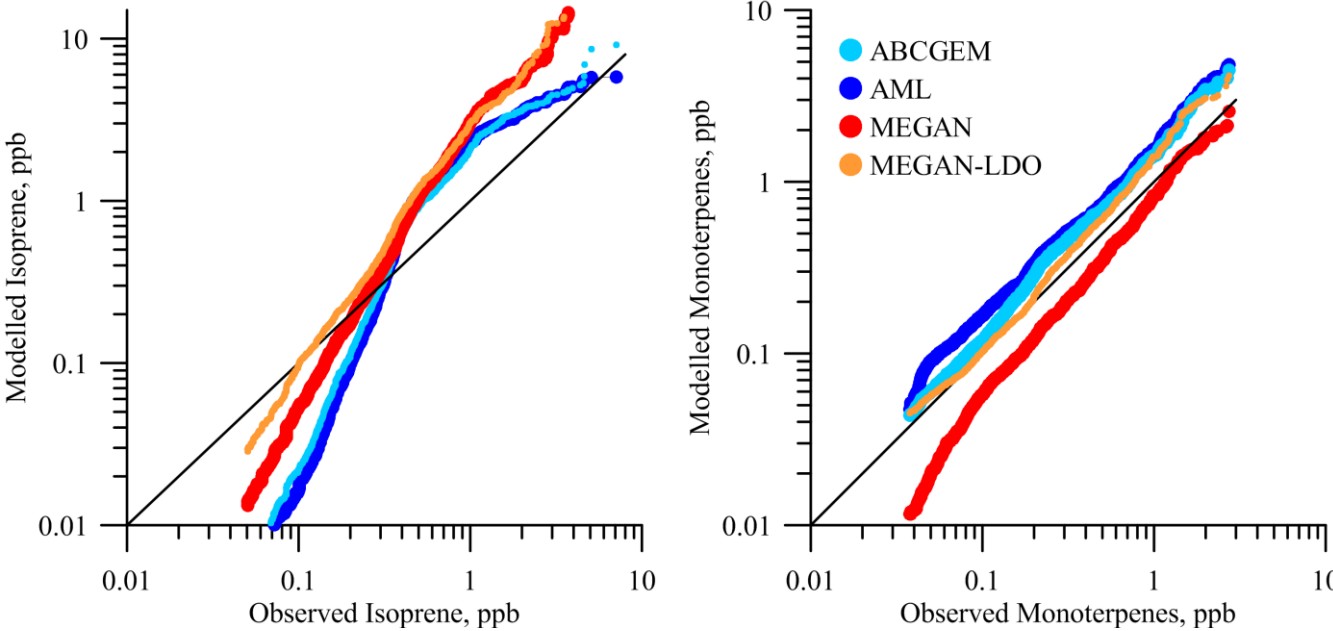

**Figure 6 Quantile-quantile plots comparing all observed data to the coincident modelled data for (left) isoprene and (right) monoterpenes. The solid line represents the 1:1 ratio. The y-axis in the isoprene plot is restricted to 15 ppb, as peak MUMBA modelled isoprene reaches 30 ppb.**

