# Peer review of "Isoprene and monoterpene emissions in south east Australia: comparison of a multi-layer canopy model with MEGAN and with atmospheric observations"

_Atmospheric Chemistry and Physics, 2017_

## Referee Comment (RC1) · Anonymous Referee #1 · 17 Nov 2017

Overview:

The paper by Emmerson and co-workers investigate the ability of two models to calculate isoprene and monoterpene biogenic emissions in Australia. Emissions calculated by the Australian Biogenic Canopy and Grass Emissions Model (ABCGEM) and by the Model of Emissions of Gases and Aerosols from Nature (MEGAN) are compared, and the total uncertainty in biogenic emissions for the Sydney Greater Metropolitan Region is estimated. Each of these biogenic emission models is then used online with the CSIRO chemistry-transport model in order to calculate isoprene and monoterpene

atmospheric concentrations to be compared with field data collected over several campaigns in Australia using a PTR-MS instrument.

This paper addresses a key question in biogenic emission modeling, with the on-going need to reduce the uncertainty associated with these emissions. In this aim, model intercomparisons and evaluations with data, such as the work presented here, definitely help to determine the strengths and weaknesses of emission schemes. The paper is therefore of true scientific interest, and is well written and clearly presented. Yet, I believe that several sections should be improved in order to clarify some of the objectives and methodologies of the work carried out, before to be published in Atmospheric Chemistry and Physics.

General comments:

The choice of the investigation strategy has to be clarified and reinforced. Indeed it is not totally obvious why in the first place the authors wouldn't build on the work presented by Emmerson et al. (2016), trying to investigate deeper the MEGAN weaknesses but would rather go for an "old" model which has not yet been published. I am convinced by the interest of this work which I am not questioning at all here but I think the reasons for such a choice should be better explained. Why is the ABCGEM so interesting for such regional applications? Is ABGCEM meant to be the model used eventually for air quality studies in Australia? Does it incorporate specificities for the region investigated? etc.

Some of the information given in the supplementary material should be moved or also given in the main core of the paper. Indeed before diving into the results, it is important to have a clear idea of the main common and different features between ABCGEM and MEGAN. This is the case of the table given in section 2 of the supplementary material. Biogenic VOCs considered in each model should also be listed in the core of the paper, together with the number of vegetation classes considered.

In section 2.1, I would enjoy reading more details regarding the campaign duration (to

better assess the representativity of data used) and the site characteristics regarding vegetation (which vegetation types? mostly vegetative surfaces or not? LAI value?).

In the last paragraph before section 4.5, typical values for the isoprene/monoterpenes ratio should be given again and the original source of this reminded. Are there any explanations for such specificity in Australia compared to other places in the world? Do we have any idea of the plant processes and sensitivity that would support such a behaviour in plant emissions?

As the light-dependency considered in MEGAN for monoterpenes is questioned, it would be interesting to have one test carried out changing such characteristic in the MEGAN model (i.e. changing the light-dependent function) to quantify the impact on ABCGEM-MEGAN discrepancies, even run as a simple test.

Specific comments:

When used, replace "inline" by "online"

Throughout the paper, "emission factor", "emission rate" and "emission flux" are several times used alternatively, while they do not represent at all the same quantity. Indeed "emission factor" represents the emission capacity of one plant species estimated in standard conditions, "emission rate" is generally used when related to an emission calculated per quantity of dry matter, and "emission flux" represent the overall quantity of compound emitted per ground surface unit, what is calculated eventually by biogenic emission models such as MEGAN or ABCGEM used here. This should be therefore corrected or clarified in the paper (for instance section 4.2 describes emission fluxes and not emission rates) and in the supplementary material.

Tables and Figures:

In figure 5, titles on the figures are particularly small and hard to read. They could be enlarged for instance without rewriting on many of them "average emission rates during SPS1".

---

## Referee Comment (RC2) · Anonymous Referee #2 · 26 Nov 2017

As the authors rightly point out, biogenic emissions play a critical role in the atmosphere and indeed the Earth system as a whole. It is therefore important that the modelling community evaluate, validate and constrain estimates of these emissions. Australia is an understudied region and one in which previous studies, including a recent one by these authors, have shown biogenic emissions models to perform poorly. The authors present the findings of a study comparing emissions estimates from two models, one developed specifically for Australian vegetation canopies, the other the widely used global MEGAN model, against observations of atmospheric concentrations of isoprene

and monoterpenes made during 5 short duration field campaigns. Such a study is much needed in order to gain critical insight into the mechanisms driving biogenic emissions and subsequent oxidation in this world region.

However I find a number of shortcomings in the present work that limit its usefulness to the atmospheric chemistry community. Chief among these are:

i. The authors are attempting to validate a model developed 15 years ago specifically for biogenic emissions from Australian ecosystems and that has clearly not been updated to reflect the current state of the art.

The ABCGEM model contains only 2 plant functional types: native trees and grasses. It is hard to believe that the rich biodiversity of Australian ecosystems can be credibly captured by such simplicity. Further, this also limits the usefulness of the model to the wider emissions modelling community.

Monoterpene emissions in the ABCGEM model are assumed purely temperature dependent while those in MEGAN are assumed partially light-dependent. The authors spend quite some time in a theoretical discussion of how this results in different activity functions in the two models and later conclude that the better performance of the ABCGEM model suggests that monoterpene emissions from native trees in Australia are less light dependent than other world regions. Given the highly complex, highly non-linear relationship between primary emissions and instantaneous atmospheric concentrations many kms away and given the ease with which this could have been tested I do not understand why the authors have not performed an additional simulation with the light-dependency switched off in MEGAN. This is incredibly straightforward and I would like to see this done before the paper is accepted.

ii. The main conclusion of the paper appears to be that biogenic emission estimates are critically dependent on the landcover maps used to model them. I feel that this is far from novel (see e.g. Guenther et al., 2006, Arneth et al., 2011 and Huang et al., 2015), and does little to help us improve emissions estimates in global models. This to my

mind is the main weakness of this study. The authors do not present any suggestions as to how we can overcome current deficiencies in a model that can be used to model estimates in any or all world regions. The authors would be well advised to consider the work presented here as a starting point. What can we learn from the apparent skill of the ABCGEM model that we can apply elsewhere?

iii. I was pleased to see the authors have explicitly included some consideration of the uncertainties associated with the emissions estimates. However, although the authors have carefully followed the error propagation methodology this considers only errors associated with measurements of a subset of the driving variables. It does not include other systemic and potentially substantial errors associated with the model parameterisations themselves (either in form or in the values of the empirical constants). As such it is rather misleading.

iv. As it stands, this is neither a rigorous evaluation of the performance of the canopy model and estimated emission rates nor is it an in-depth analysis of the atmospheric chemistry in the region. In fact, I find it hard to understand what the authors intend. It seems mostly to be a lengthy appraisal of model treatment of leaf area index (LAI) and the deficiencies of the various available landcover maps. As such, it does not to my mind fit within the remit of ACP but would be far better placed in sister journal GMD, although with the above caveats regarding the need for better evaluation of the canopy model itself.

In its present form, I do not consider this work to be suitable for publication in ACP. I would suggest that as a bare minimum, the authors need to address my concerns above and to reverse their current approach and concentrate on spatial and temporal distributions of modelled isoprene and monoterpene emissions, and modelled and measured atmospheric concentrations which are of far greater interest to the wider atmospheric science community (the primary audience of ACP) and which offer the possibility of real advances in the field. Further, I would also like to see how modelled concentrations of primary oxidants and oxidation products compare with those measured at the various sites. Given the highly complex and highly non-linear chemistry of bVOCs comparison of concentrations cannot necessarily be used to deduce skill in modelling emissions. Further validation would be useful.

At present this paper is being used as a vehicle for a description of the ABCGEM model which seems to have little applicability outside of this region. If that is the intention of the authors I would recommend seeking publication in GMD with substantially more consideration given to how the comparison (with MEGAN output and observations) can be applied to improve model performance. AND to take steps to do just that with further sensitivity tests, e.g. MEGAN without light-dependent monoterpene emissions.

Specific comments: Overall I feel that much of the discussion and presentation of data comparing the functional form of the activity factors in ABCGEM and MEGAN would be better placed in the SI. Likewise the exhaustive coverage of LAI, which seems to be over-accounted for in ABCGEM.

I would like to see far greater detail of the measurement sites, their footprint and dominant air mass origin during the period of the campaign(s), vegetation / ecosystem type, etc included in the main paper.

The authors use the term "emission factor" to mean several different things at various times during the analysis, in particular in Section 4 where they continue to use the same phrase to describe both a basal emission factor (i.e. emission rate at standard conditions) and a landscape emission factor (i.e. some kind of average gridcell BEF which accounts for contributions for all vegetation types within a model gridcell). I think it is this phrasing that makes this section so hard to follow when the authors introduce the effect of differences in LAI between MEGAN and ABCGEM.

Throughout: "inline" - do the authors mean "online"?

Abstract:

L23: surely "simpler" rather than "simplified" as there is no suggestion that the authors

have reduced ABCGEM in any way for this work.

Introduction:

Throughout: there are far too many unsubstantiated statements made without reference to supporting literature, in particular:

L15: Please supply a reference for the C-CTM when first introduced.

L21-22: Do the authors have evidence that these forests have a substantial impact?

L31-32: How is this relevant in a region where isoprene:monoterpene is unity?

Methods: 2.1 As stated previously I would like to see far greater detail of the campaigns, the sites and meteorological conditions, the measurements available, etc. I consider this essential for the Brinsgelly and Randwick data which have not been previously published.

2.2 As this model has not previously been described further detail is required in the main text. It would be far easier to follow if the authors presented the parameterisations here rather than attempt to describe in words.

2.2.1 Why set B to a value of 400 g m-2? This does not seem consistent with the description in Section 1 of the SI.

Also why devote so much of this section to a description of the grass PFT when it is promptly left out of the model?

2.2.2 This section is a discussion not a method. Further, as the authors devote so much time discussing the light-dependence of monoterpene emissions it would be useful to learn whether previous experiments on native Australian vegetation have shown evidence either way.

Section 3: Why is the C-CTM not described as part of the Methods section like ABCGEM is?

**ACPD**

L30: I suggest showing the model domains on Fig. 1.

3.1 I find this description of the seeming over dependence of ABCGEM on LAI extremely difficult to follow. However, if it is driven with 1980s LAI as a default it is good to see that the authors have conducted the AML sensitivity test.

L25 Typo: "ccalculated"

Section 4 Results and discussion

4.1: As stated above I found this particularly hard to follow in part because of the use of "emission factor" to describe several things.

L14-23: I don't think that the "emission factors" being compared here are directly comparable. . .

4.2 L8: See point below regarding Figure 4.

L10-11: There would be no reason for anyone to expect the emissions to scale linearly with LAI . . . This comes back again to the confusing and inconsistent terminology used with regard to emission factors and emission rates.

L20-21: The issue of whether or not monoterpene emissions should be treated as light-dependent or not is really quite an important one as it is a fundamental mechanism rather than a model "artefact" such as LAI. Yet the authors seem to have made no attempt to investigate this further. To my mind this is where the real novelty could lie, and where the authors could offer something to the modelling community as a whole. Is it the case that MEGAN is incorrect to assume that all monoterpene emissions have some light-dependence? Should these factors be PFT-specific? etc.

L25-26: Again, surely this is to be expected . . .

Section 4.3 appears to be missing

4.4 L3-10: It would be helpful for the authors to include % differences to put their

absolute changes into context.

4.5 Overall I find the analyses of the output of the two model with respect to observations well done. However, for this paper to be suitable for ACP I feel that it is this section that should form the focus of the paper rather than the preceding consideration of LAI.

p9, L3-6: Agreed. If updating a model developed 15 years ago to use a more up-to-date input dataset causes the model to "fail" the model clearly needs further development which is one of my key concerns with this work as it is presented. Why use this model rather than using the wealth of observations to improve the skill of MEGAN for this region? The two models take essentially the same approach to estimating emissions (i.e. empirical rather than mechanistic) so it is not evident what we gain from going back to the older model.

L14-15: I feel the wind roses should be in the main paper as this consideration of emissions vs meteorology / chemistry is important.

L20-25: MEGAN appears to have a good fit to the profile of monoterpene concentrations at Randwick and (to a lesser extent) SPS1 whereas at Bringelly all 3 simulations vastly over predict night-time concentrations. It seems far from clear that the light-dependence of monoterpene emissions is the only issue here. I suggest the authors need to investigate more fully.

p10, L1-5: I'm not sure that the q-q plots add much to the discussion and would suggest they be moved to the Supplementary.

L19-24: See previous comments regarding the estimated uncertainty.

5 Conclusions L25: I wouldn't consider the two models to have independent approaches to estimating emissions; both are based on the Guenther empirical algorithms developed in the 1990s.

p11, L21-23: I would argue that in spite of using the same chemistry scheme some

of the difference in concentrations between the two models will still be due to chemical processing as it is highly non-linear and strongly dependent on VOC-NOx ratios which will differ if VOC emissions change. I agree that the predominant cause of the differences are differences in emissions.

L31: As noted previously I feel the authors need to give far more detail of these two sets of measurements as they have not been previously published.

p12, L7-8: Agreed, but that doesn't preclude the authors from investigating further at this stage.

Table 2. What about PAR and Temperature for each campaign? (average and some measure of range)

Figures 2 -4 would be better presented in the SI (but see below)

Figures 3 & 4 I feel the way the data is presented is fundamentally flawed. LAI is not a discrete variable but rather a weighted average for a grid cell based on proportional land cover. It therefore makes no sense to plot the data showing ranges for emissions but not LAI. While I understand that the authors have binned the data by a range of LAI it is still not appropriate to plot the emission rate against the mid-point of the LAI bin. At the very least, it should be a weighted average of the LAIs of the grid cells within that bin but even then I would question its appropriateness.

---

## Author Comment (AC1) · 14 Feb 2018

> Overview: The paper by Emmerson and co-workers investigate the ability of two models to calculate isoprene and monoterpene biogenic emissions in Australia. Emissions calculated by the Australian Biogenic Canopy and Grass Emissions Model (ABCGEM) and by the Model of Emissions of Gases and Aerosols from Nature (MEGAN) are compared, and the total uncertainty in biogenic emissions for the Sydney Greater Metropolitan Region is estimated. Each of these biogenic emission models is then used online with the CSIRO chemistry-transport model in order to calculate isoprene and monoter-

pene atmospheric concentrations to be compared with field data collected over several campaigns in Australia using a PTR-MS instrument.

This paper addresses a key question in biogenic emission modeling, with the on-going need to reduce the uncertainty associated with these emissions. In this aim, model intercomparisons and evaluations with data, such as the work presented here, definitely help to determine the strengths and weaknesses of emission schemes. The paper is therefore of true scientific interest, and is well written and clearly presented. Yet, I believe that several sections should be improved in order to clarify some of the objectives and methodologies of the work carried out, before to be published in Atmospheric Chemistry and Physics.

Response: We thank the reviewer for this overview. We have made significant changes to the manuscript based on both reviewers' helpful comments.

> General comments: The choice of the investigation strategy has to be clarified and reinforced. Indeed it is not totally obvious why in the first place the authors wouldn't build on the work presented by Emmerson et al. (2016), trying to investigate deeper the MEGAN weaknesses but would rather go for an "old" model which has not yet been published. I am convinced by the interest of this work which I am not questioning at all here but I think the reasons for such a choice should be better explained. Why is the ABCGEM so interesting for such regional applications? Is ABCGEM meant to be the model used eventually for air quality studies in Australia? Does it incorporate specificities for the region investigated? etc.

Response: This comparison between the current world class global model, MEGAN, and an older regionally developed model, ABCGEM, was undertaken because of the following:

1) There have been very few experimental studies of VOC emissions from vegetation in Australia.

[Figure]

2) The VOC emissions from Australian vegetation may be different in magnitude and behaviour from those studied in the northern temperate regions and in the tropics because Australian vegetation was isolated from other regions for many tens of millions of years and in general adapted to infertile deeply weathered ancient soils and a regime of intense fires (Orians and Milewski, 2007), factors that could affect the evolutionary choices concerning plant VOC emissions.

3) The study of Emmerson et al (2016) indicated significant differences between VOC levels modelled using MEGAN and those observed for SE Australia.

4) In studies of modelling of complex systems such as climate or hydrology, there is empirical evidence that the total knowledge about the system is not held exclusively by the world leading model, but rather the best results are derived from an ensemble of models. We presume the same phenomena applies to models of VOC emissions from vegetation.

5) Comparison of such models of a complex system can provide useful scientific insights.

Because an older regionally developed model, ABCGEM, was already available we hypothesised that an efficient way to identify some of the limitations and strengths of both the input data and modelling of emissions of VOCs from vegetation in Australia would be by comparison of MEGAN with ABCGEM. This has been undertaken. This work is of significance to understanding the global atmosphere as Australia is one of the four continents in the Southern Hemisphere and its VOC emissions will significantly affect the levels of Southern Hemisphere VOCs and SOA.

Text changes: we have rewritten 2 paragraphs in the introduction (page 2 paragraphs 3 and 4) to read:

"The south east coastal ecosystem of Australia is dominated by eucalypt trees, and is identified as a global BVOC emitting hotspot (Guenther et al., 2006). However recent work by Emmerson et al. (2016) demonstrated considerable discrepancies using MEGAN when compared to atmospheric observations over south-eastern Australia. Emmerson et al. (2016) postulated that the discrepancies calculated by MEGAN in Australia were due to unrepresentative emission factors, the majority coming from studies both in Australia and overseas on eucalypt saplings under laboratory conditions. The VOC emissions from Australian vegetation may be different in magnitude and behaviour from those studied in the northern temperate regions and in the tropics because Australian vegetation was isolated from other regions for many tens of millions of years and in general adapted to infertile deeply weathered ancient soils and a regime of intense fires (Orians and Milewski, 2007), factors that could affect the evolutionary biology of plant VOC emissions (Fernández-Martínez et al., 2017).

Here we use further modelling and comparisons with atmospheric observations to try to understand why MEGAN performs poorly over south-eastern Australia. A comparison of MEGAN with the unpublished locally developed Australian Biogenic Canopy and Grass Emissions Model (ABCGEM) could provide useful scientific insights into the cause of the MEGAN discrepancy in SE Australia. This is a region with very few experimental studies of BVOCs, and comparison with ABCGEM results may be an efficient way to identify the limitations and strengths of MEGAN for South-eastern Australia."

> Some of the information given in the supplementary material should be moved or also given in the main core of the paper. Indeed before diving into the results, it is important to have a clear idea of the main common and different features between ABCGEM and MEGAN. This is the case of the table given in section 2 of the supplementary material.

Response: Table 2 has been moved back into the main paper.

Page 5, line 8-9 "Differences in the inputs required by each model are given below and in Table 1"

> Biogenic VOCs considered in each model should also be listed in the core of the paper, together with the number of vegetation classes considered.

Response: This information is in the paper but perhaps not clearly presented. We will modify the Section 2 Methods to correct for this.

Page 3 line 5, add "All measurements of monoterpenes by PTRMS are of the combined species at mass to charge ratio m/z = 137."

Page 3 line 38, add "All calculations of monoterpenes in ABCGEM are on the lumped species."

Page 4 line 39, insert "CB05 combines individual monoterpenes into one lumped monoterpene species."

Page 5 line 11, add "The vegetation class used in ABCGEM is eucalypt forest, with the proviso that the canopy height and LAI are independent variables.

Page 5 line 32, add "The vegetation classes used in MEGAN are embedded within plant functional types and emission factor maps as described in Emmerson et al. (2016)."

> In section 2.1, I would enjoy reading more details regarding the campaign duration (to better assess the representativity of data used) and the site characteristics regarding vegetation (which vegetation types? mostly vegetative surfaces or not? LAI value?).

Response: The LAI values from both datasets have been extracted and further details of the surrounding vegetation have been included. The data citations have been moved to the data provision section. Page 3 paragraph 1 has been changed to read:

"Figure 1 shows the locations of the five field campaigns conducted within the Sydney GMR, The Sydney Particle Studies SPS1 and SPS2, Measurements of Urban Marine and Biogenic Air (MUMBA), and campaigns at Bringelly and Randwick. Each campaign measured hourly concentrations of isoprene and monoterpenes using the same PTR-MS instrument and employed standard calibration gases. Observations of monoterpenes by PTR-MS are based on the calibration and measurement of the combined monoterpene species at mass to charge ratio m/z = 81 for the Bringelly and Randwick campaigns and at mass to charge ratio m/z = 137 for the later SPS1, SPS2 and

MUMBA campaigns. The change was made to improve sensitivity and reduce potential interferences. Three of the campaigns were documented in Emmerson et al. (2016): SPS1 and SPS2 were located at Westmead, a suburban site 21 km west of Sydney (150.9961°E, 33.8014°S). SPS1 ran from 18 February – 7 March 2011, and SPS2 from 14 April – 14 May 2012, (Cope et al., 2014). The Westmead site is located next to a grass playing field within hospital grounds, with a line of trees to the west and south, separating the site from trains, roads and housing beyond. The MODIS LAI value for Westmead is 1.2 m2 m-2. Dunne et al. (2018) have shown night time interference from wood smoke compounds in the isoprene signal taken during SPS2. Therefore the SPS2 isoprene observational dataset is restricted to daylight hours between 9am and 6pm. MUMBA was situated near the coast at Wollongong, (150.8995°E, 34.3972°S) from 22 December 2012 – 15 February 2013 (Paton-Walsh et al., 2017). The MUMBA site is also grassy (LAI of 1.7 m2 m-2), separated from the ocean 0.5 km to the east by a strip of eucalypt trees. A 400 m eucalypt forested escarpment is 3 km to the west.

A suite of meteorological data, including wind speed and direction were taken at each of the field campaign sites, with details given in the indicated literature. Polar bivariate plots are also shown in Figure 1 which give observed isoprene volume mixing ratios by wind speed and direction at each of the campaign sites. These show that the peak isoprene measurements are not always associated with the dominant wind directions, but are correlated with the directions of the forested regions to the northwest and west of each of the sites."

Page 3 paragraph 2 has been changed to read:

"PTR-MS observations were undertaken in summer 2007 at Bringelly, a semi-rural site (150.7619°E, 33.9177°S, 24 January – 27 February 2007), and Randwick, 8 km from Sydney centre (151.2428°E, 33.9318°S, 28 February – 19 March 2007). Both sites are air quality management stations operated by the NSW government and take wind speed and direction, temperature and relative humidity measurements, along with ozone, NOx and particulate matter

(www.environment.nsw.gov.au/AQMS/SiteSyd.htm). The inlet height for the PTR-MS instrument was approximately 4.5 m at both sites. Bringelly is located on reserve of open grassed council land (LAI of 2.1 m2 m-2), with occasional trees and bordered by Ramsay road at 53 m elevation. Low density housing is to the east. The heavily eucalypt forested Blue Mountains are 16 km to the west, which is where the source of the observed isoprene comes from. However the predominant wind directions are from the south-west and east.

The Randwick station at 28 m elevation is sited on a grassland paddock within army barracks, bordered by trees. The barracks are within a housing suburb (LAI of 0.5 m2 m-2). The dominant wind direction is from the south, with the dominant BVOC source coming from the north-west, consistent with the SPS1 BVOC source direction."

> In the last paragraph before section 4.5, typical values for the isoprene/monoterpenes ratio should be given again and the original source of this reminded.

Page 8 line 27 insert "Emmerson et al (2016) found ratios close to 1 for observed levels in the Sydney basin. This is in contrast to a ratio of 0.18 found in boreal forests dominated by monoterpenes (Spirig et al., 2004), and to a ratio of 26.4 in deciduous Michigan forests dominated by isoprene (Kanawade et al., 2011)."

> Are there any explanations for such specificity in Australia compared to other places in the world?

We searched the literature for the reasons why plants emit BVOCs, and found two good references in Harrison et al (2013) and Fernandez-Martinez et al (2017). The Harrison paper collated emissions data from many sources suggesting that most plants emit either isoprene or monoterpenes, and of those emitting both there is a trade-off favouring one BVOC over the other. Whilst eucalypts are not mentioned specifically, they are in a minority group of plants emitting both BVOC species strongly. Other processes such as leaf age and leaf area were investigated, in general showing isoprene negatively correlated with leaf age and positively correlated with leaf area. Eucalypts are not deciduous therefore the leaf age is longer, however the plant could invest energy over the long term to produce monoterpenes as a defence mechanism against herbivory.

Fernandez-Martinez et al investigate the relationship between nutrient availability and BVOC emissions, suggesting that plants able to store monoterpenes (such as eucalypts) were associated with poor nutrient availability. Australian soils fit this category.

Page 11 line 9-23 marked up copy- inserted:

"This carbon ratio is most likely controlled by metabolic processes within the plants and as such is a valid test of the models. The biochemistry behind this competition is explained in Harrison et al. (2013) who present emission capacities from species worldwide emitting both isoprene and monoterpene. Two thirds of the 80 cases have ratios greater than 1. Monoterpene emissions are favoured in nitrogen poor conditions (Fernández-Martínez et al., 2017) in species with a long leaf lifespan (Harrison et al. 2013), conditions matching Australia."

> Do we have any idea of the plant processes and sensitivity that would support such a behaviour in plant emissions?

Response: see above response, and also our first response. We suggest that Australian vegetation has evolved independently from northern hemisphere species, the landmass being isolated from other regions for many tens of millions of years and in general adapted to infertile deeply weathered ancient soils and a regime of intense fires (Orians and Milewski, 2007), factors that could affect the evolutionary biology of plant VOC emissions (Fernández-Martínez et al., 2017). This paragraph has been inserted into the introduction.

> As the light-dependency considered in MEGAN for monoterpenes is questioned, it would be interesting to have one test carried out changing such characteristic in the MEGAN model (i.e. changing the light-dependent function) to quantify the impact on ABCGEM-MEGAN discrepancies, even run as a simple test.

The light dependence of monoterpenes has been switched off in the MEGAN-LDO test for all the campaign periods studied. The results are so interesting that they have been threaded throughout the paper, making this a major focus of the revised paper.

In summary, switching of the light dependence of monoterpenes increased the night time (baseline) emission flux by 90 – 100% on the original MEGAN run. These higher emissions increase the modelled monoterpenes at night by a factor of 2 -3 improving the comparisons with observations considerably. The emissions during the day were not impacted as much because other activity functions in MEGAN reduce the emissions, but strong chemical removal processes during the day mean that the diurnal time series of monoterpenes for ABCGEM, AML and MEGAN-LDO were similar. Whilst the MEGAN-LDO test does not impact on the emissions of isoprene, the change in oxidant chemistry due to the increased monoterpenes has changed the isoprene volume mixing ratios, in most cases reducing them by 4% during the daytime and improving the comparison with observations. Overall switching of the light dependence of monoterpenes has reduced the bias in the MEGAN model and improved the carbon ratio towards the observations.

> Specific comments: When used, replace "inline" by "online" Response: Done.

> Throughout the paper, "emission factor", "emission rate" and "emission flux" are several times used alternatively, while they do not represent at all the same quantity. Indeed "emission factor" represents the emission capacity of one plant species estimated in standard conditions, "emission rate" is generally used when related to an emission calculated per quantity of dry matter, and "emission flux" represent the overall quantity of compound emitted per ground surface unit, what is calculated eventually by biogenic emission models such as MEGAN or ABCGEM used here. This should be therefore corrected or clarified in the paper (for instance section 4.2 describes emission fluxes and not emission rates) and in the supplementary material.

We thank this reviewer for taking the time to explain this. We have standardised the use

of "emission factor", "emission rate" and "emission flux" throughout the paper according to the definitions above.

> Tables and Figures: In figure 5, titles on the figures are particularly small and hard to read. They could be enlarged for instance without rewriting on many of them "average emission rates during SPS1".

Response: Done

References

Cope, M., Keywood, M., Emmerson, K., Galbally, I., Boast, K., Chambers, S., Cheng, M., Crumeyrolle, S., Dunne, E., Fedele, F., Gillett, R. W., Griffiths, A., Harnwell, J., Katzfey, J., Hess, D., Lawson, S., Miljevic, B., Molloy, S., Powell, J., Reisen, F., Ristovski, Z., Selleck, P., Ward, J., Zhang, C., and Seng, J.: The Sydney Particle Study. CSIRO, Australia. Available at http://www.environment.nsw.gov.au/aqms/sydparticlestudy.htm, 2014.

Dunne, E., Galbally, I. E., Cheng, M., Selleck, P., Molloy, S. B., and Lawson, S. J.: Comparison of VOC measurements made by PTR-MS, Adsorbent Tube/GC-FID-MS and DNPH-derivatization/HPLC during the Sydney Particle Study, 2012: a contribution to the assessment of uncertainty in current atmospheric VOC measurements, Atmos. Meas. Tech., 11, 141-159, https://doi.org/10.5194/amt-11-141-2018, 2018.

Emmerson, K. M., Galbally, I. E., Guenther, A. B., Paton-Walsh, C., Guerette, E. A., Cope, M. E., Keywood, M. D., Lawson, S. J., Molloy, S. B., Dunne, E., Thatcher, M., Karl, T., and Maleknia, S. D.: Current estimates of biogenic emissions from eucalypts uncertain for southeast Australia, Atmos Chem Phys, 16, 6997-7011, 10.5194/acp-16-6997-2016, 2016.

Fernández-Martínez, M., Llusià, J., Filella, I., Niinemets, Ü., Arneth, A., Wright, I. J., Loreto, F., and Peñuelas, J.: Nutrient-rich plants emit a less intense blend of volatile isoprenoids, New Phytol, doi: 10.1111/nph.14889, 2017.

Harrison, S. P., Morfopoulos, C., Dani, K. G. S., Prentice, I. C., Arneth, A., Atwell, B. J., Barkley, M. P., Leishman, M. R., Loreto, F., Medlyn, B. E., Niinemets, U., Possell, M., Penuelas, J., and Wright, I. J.: Volatile isoprenoid emissions from plastid to planet, New Phytol, 197, 49-57, 10.1111/nph.12021, 2013.

Kanawade, V. P., Jobson, B. T., Guenther, A. B., Erupe, M. E., Pressley, S. N., Tripathi, S. N., and Lee, S. H.: Isoprene suppression of new particle formation in a mixed deciduous forest, Atmos Chem Phys, 11, 6013-6027, 10.5194/acp-11-6013-2011, 2011.

Orians, G. H., and Milewski, A. V.: Ecology of Australia: the effects of nutrient-poor soils and intense fires, Biol Rev, 82, 393-423, 10.1111/j.1469-185X.2007.00017.x, 2007.

Spirig, C., Guenther, A., Greenberg, J. P., Calanca, P., and Tarvainen, V.: Tethered balloon measurements of biogenic volatile organic compounds at a Boreal forest site, Atmos Chem Phys, 4, 215-229, 2004.

———————————————

---

## Author Comment (AC2) · 14 Feb 2018

As the authors rightly point out, biogenic emissions play a critical role in the atmosphere and indeed the Earth system as a whole. It is therefore important that the modelling community evaluate, validate and constrain estimates of these emissions. Australia is an understudied region and one in which previous studies, including a recent one by these authors, have shown biogenic emissions models to perform poorly. The authors present the findings of a study comparing emissions estimates from two models, one developed specifically for Australian vegetation canopies, the other the widely used

global MEGAN model, against observations of atmospheric concentrations of isoprene and monoterpenes made during 5 short duration field campaigns. Such a study is much needed in order to gain critical insight into the mechanisms driving biogenic emissions and subsequent oxidation in this world region.

Response: We thank the reviewer for this overview. We have made significant changes to the manuscript based on both reviewers' helpful comments. Because the referee has made a number of related comments, we number them for cross-reference here.

However I find a number of shortcomings in the present work that limit its usefulness to the atmospheric chemistry community. Chief among these are:

1. The authors are attempting to validate a model developed 15 years ago specifically for biogenic emissions from Australian ecosystems and that has clearly not been updated to reflect the current state of the art.

Response: We are not trying to validate ABCGEM. A limitation of the discussion (ACPD) version of this manuscript was the inadequate presentation of the reason for undertaking this model comparison. As described also to Anonymous Referee#1 the following is the explanation. This comparison between the current world class global model, MEGAN, and an older regionally developed model, ABCGEM, was undertaken because of the following:

a) There have been very few experimental studies of VOC emissions from vegetation in Australia.

b) The VOC emissions from Australian vegetation may be different in magnitude and behaviour from those studied in the northern temperate regions and in the tropics because Australian vegetation was isolated from other regions for many tens of millions of years and, in general, adapted to infertile deeply weathered ancient soils and a regime of intense fires (Orians and Milewski, 2007), factors that could affect the evolutionary choices concerning plant VOC emissions (Fernández-Martinez et al. 2017).

c) The study of Emmerson et al (2016) indicated significant differences between VOC volume mixing ratios modelled using MEGAN and those observed for SE Australia.

d) In studies of modelling of complex systems such as climate or hydrology, there is empirical evidence that the total knowledge about the system is not held exclusively by the world leading model, but rather the best results are derived from an ensemble of models. We presume the same phenomena applies to models of VOC emissions from vegetation.

e) Comparison of such models of a complex system can provide useful scientific insights. An older regionally developed model, ABCGEM, was already available and we hypothesised that an efficient way to identify some of the limitations and strengths of both the input data and modelling of emissions of VOCs from vegetation in Australia would be by comparison of MEGAN with ABCGEM. This has been undertaken. This work is of significance to understanding the global atmosphere as Australia is one of the four continents in the Southern Hemisphere and its VOC emissions will significantly affect the levels of Southern Hemisphere VOCs and SOA.

To reiterate: Our desire is not to advance ABCGEM, it is to learn about the limitations and strengths of both the input data and modelling of emissions of VOCs from vegetation in Australia.

2. The ABCGEM model contains only 2 plant functional types: native trees and grasses. It is hard to believe that the rich biodiversity of Australian ecosystems can be credibly captured by such simplicity.

Response: ABCGEM has functioned well in describing emissions in a moist temperate coastal system in Australia, a system compatible with the available emission studies of Australian vegetation. We have not suggested that ABCGEM, as presented, would capture with small uncertainty the variability of VOC emissions from rich biodiversity of Australian ecosystems.

3. Further, this also limits the usefulness of the model to the wider emissions modelling community.

Response: We are not recommending ABCGEM is used by the wider modelling community. We are using it for the reasons given previously, to learn about the limitations and strengths of both the input data and modelling of emissions of VOCs from vegetation in Australia.

4. Monoterpene emissions in the ABCGEM model are assumed purely temperature dependent while those in MEGAN are assumed partially light-dependent. The authors spend quite some time in a theoretical discussion of how this results in different activity functions in the two models and later conclude that the better performance of the ABCGEM model suggests that monoterpene emissions from native trees in Australia are less light dependent than other world regions. Given the highly complex, highly non-linear relationship between primary emissions and instantaneous atmospheric concentrations many kms away and given the ease with which this could have been tested I do not understand why the authors have not performed an additional simulation with the light-dependency switched off in MEGAN. This is incredibly straightforward and I would like to see this done before the paper is accepted.

Response: We thank the Referee for this recommendation and have performed a modelling run with and without light dependency (referred to as MEGAN-LDO) and this is now included in the paper. The results of this are so interesting it has become a 4th sensitivity run and the paper has been re-written to include it. The emissions of monoterpenes in MEGAN have increased by 90 - 100% at night time when the boundary layer is low and the chemical removal processes are slow. This results in a doubling of the night time monoterpenes at each of the field sites, improving the comparison with the observations. Whilst the MEGAN-LDO test does not impact on the emissions of isoprene, the change in oxidant chemistry due to the increased monoterpenes has changed the isoprene volume mixing ratios, in most cases reducing them by 4% during the daytime and improving the comparison with observations. Overall switching of the

light dependence of monoterpenes has reduced the bias in the MEGAN model and improved the carbon ratio towards the observations.

5. ii. The main conclusion of the paper appears to be that biogenic emission estimates are critically dependent on the landcover maps used to model them. I feel that this is far from novel (see e.g. Guenther et al., 2006, Arneth et al., 2011 and Huang et al., 2015), and does little to help us improve emissions estimates in global models. This to my mind is the main weakness of this study.

Response: There is little previous work on VOC emissions on the continent of Australia. There are several significant conclusions. The others concern:

a) The quantification of the uncertainty estimates, both top-down and bottom-up, in BVOC emissions from south-eastern Australia.

b) The influence of differences in the activity functions on the agreement of model-model and model-observation comparisons with both the temperature (for isoprene) and light dependence (for monoterpenes).of the comparisons.

c) The observed isoprene to monoterpene carbon emission ratio and its context.

One of the findings of this comparison (not immediately evident from the previous work (Emmerson et al. 2016) is that, for this region of SE Australia, biogenic emission estimates are critically dependent on the landcover maps used to model them and the available maps have significant deficiencies. We see what the referee calls "the major weakness of the paper", as one component of its strength: that is the lesson that when modelling VOC emissions from hitherto poorly explored regions of the world (from the perspective of VOC emissions) critical attention needs to be given to verifying the underlying input data.

6. The authors do not present any suggestions as to how we can overcome current deficiencies in a model that can be used to model estimates in any or all world regions. The authors would be well advised to consider the work presented here as a starting

point. What can we learn from the apparent skill of the ABCGEM model that we can apply elsewhere?

Response: We are not suggesting that the results from the ABCGEM study can be used elsewhere. The paper highlights the fact that estimates of VOC emissions from vegetation in SE Australia will not be improved without further experimental studies of emissions and atmospheric concentrations in the region. Thanks to reviewer suggestions, we have completed the model runs whereby the light dependence of monoterpene species has been switched off, and this has shown that the issue of whether light dependence is applicable to Australian vegetation is central to addressing the discrepancies calculated by MEGAN. As explained in response 30, future versions of MEGAN will use landscape specific light dependent parameters.

7. iii. I was pleased to see the authors have explicitly included some consideration of the uncertainties associated with the emissions estimates. However, although the authors have carefully followed the error propagation methodology this considers only errors associated with measurements of a subset of the driving variables. It does not include other systemic and potentially substantial errors associated with the model parameterisations themselves (either in form or in the values of the empirical constants). As such it is rather misleading.

Response: We thank the Referee for their comments on the uncertainty analysis and the implied requirement for an overall uncertainty analysis. One of the outcomes in the existing paper of the comparison of the VOC emission estimates of AGCGEM and MEGAN is an uncertainty estimate that includes the wider terms as described by the Referee. We consider that there are two assessments of uncertainty, the bottom-up one described above and the top-down assessment available by comparing the two models and then the models with observations via C-CTM. This is reported on page 13 lines 25 to 33 of the marked up paper.

"One goal in this work is to calculate a total uncertainty in BVOC emissions for the

Sydney GMR. Two approaches are used in this paper. In section 2.3.1 a bottom up uncertainty assessment for ABCGEM (presented in the Supplementary Material) was discussed. Here a top-down assessment is made utilizing the calculated normalised mean biases between the models and observations in Table 3. These provide the scatter from model to model and campaign to campaign as a measure of uncertainty. The 95% confidence limits from the NMBs in Table 3 are equivalent to uncertainties of factors of ~2 for isoprene and ~3 for monoterpenes. This is consistent with the estimate of a factor of 2 from the bottom up estimate that omits uncertainty due to knowledge missing from the models, and also consistent with the factors of 4 difference in the modelled carbon rations between ABCGEM and MEGAN."

8. iv. As it stands, this is neither a rigorous evaluation of the performance of the canopy model and estimated emission rates nor is it an in-depth analysis of the atmospheric chemistry in the region. In fact, I find it hard to understand what the authors intend. It seems mostly to be a lengthy appraisal of model treatment of leaf area index (LAI) and the deficiencies of the various available landcover maps. As such, it does not to my mind fit within the remit of ACP but would be far better placed in sister journal GMD, although with the above caveats regarding the need for better evaluation of the canopy model itself. In its present form, I do not consider this work to be suitable for publication in ACP.I would suggest that as a bare minimum, the authors need to address my concerns above and to reverse their current approach and concentrate on spatial and temporal distributions of modelled isoprene and monoterpene emissions, and modelled and measured atmospheric concentrations which are of far greater interest to the wider atmospheric science community (the primary audience of ACP) and which offer the possibility of real advances in the field. Further, I would also like to see how modelled concentrations of primary oxidants and oxidation products compare with those measured at the various sites. Given the highly complex and highly non-linear chemistry of bVOCs comparison of concentrations cannot necessarily be used to deduce skill in modelling emissions. Further validation would be useful.

Response: The paper nowhere has the purpose of being "an in-depth analysis of the atmospheric chemistry in the region." It is unfair to judge it against that. Note that in all the model runs all other pollutant emissions, atmospheric concentrations, physical and meteorological conditions are identical, that is the purpose of using a single modelling framework the C-CTM.

The "highly complex and highly non-linear chemistry of bVOCs" is a second order effect here as (1) the paper deals with the bVOC atmospheric concentrations which are a function of their emissions and initial loss mechanisms and (2) the observed and modelled concentrations, Figure 7, are relatively low with 93% below 1 ppb, levels unlikely to drive a highly non-linear chemistry. Comparisons of O3, NOx and the ratio of isoprene to isoprene products are made in our previous paper, Emmerson et al. (2016). Unfortunately there are no measurements of OH, HO2 and NO3 on the Australian mainland with which to make any assessments.

The purpose of the paper, as explained in response to Comment 1, is to it is to learn about the limitations and strengths of both the input data and modelling of emissions of VOCs from vegetation in Australia. This work is of significance to understanding the global atmosphere as Australia is one of the four continents in the Southern Hemisphere and its VOC emissions will significantly affect the levels of Southern Hemisphere VOCs and SOA.

We have taken on board the comments regarding the interest of the work to ACP readers and have replaced one of the comparisons with LAI (figure 4) with temporal time series of the domain average isoprene and monoterpene emission fluxes. Many of the conclusions regarding the differences between MEGAN and ABCGEM remain the same in that MEGAN isoprene emissions are 2-3 times higher than ABCGEM, that changing the LAI dataset in the AML test has had minor (10-20%) impacts on isoprene and monoterpene emissions, and that ABCGEM daytime emissions of monoterpenes are ~2 times higher than MEGAN. The MEGAN-LDO test has brought night time emissions of monoterpenes to be very similar to those of ABCGEM.

It is worth noting that this is only the second paper to examine *in detail* the spatial and temporal distributions of modelled VOC emissions from vegetation in SE Australia. Neither the experimental base, nor the background of multiple prior studies exist to produce greater in-depth analysis.

9. At present this paper is being used as a vehicle for a description of the ABCGEM model which seems to have little applicability outside of this region. If that is the intention of the authors I would recommend seeking publication in GMD with substantially more consideration given to how the comparison (with MEGAN output and observations) can be applied to improve model performance. AND to take steps to do just that with further sensitivity tests, e.g. MEGAN without light-dependent monoterpene emissions.

Response: See responses 1,5, 7 and 8.

Specific comments:

10. Overall I feel that much of the discussion and presentation of data comparing the functional form of the activity factors in ABCGEM and MEGAN would be better placed in the SI. Likewise the exhaustive coverage of LAI, which seems to be over-accounted for in ABCGEM.

Response: We have moved the plots of the activity functions to the supplementary section, but have retained the text on the different model treatments of monoterpenes which are now included in the model description sections.

As mentioned in response 8, we have taken out section 4.2 comparing the emission fluxes with LAI and replaced the figure and section with domain average temporal plots of emission fluxes for each campaign period.

11. I would like to see far greater detail of the measurement sites, their footprint and dominant air mass origin during the period of the campaign(s), vegetation / ecosystem type, etc included in the main paper.

Response: This has been done see text below from p4 of revised paper. We have also added polar bivariate plots of observed isoprene at each site in figure 1, which combines a wind rose with the dominant isoprene source locations.

"Figure 1 shows the locations of the five field campaigns conducted within the Sydney GMR, The Sydney Particle Studies SPS1 and SPS2, Measurements of Urban Marine and Biogenic Air (MUMBA), and campaigns at Bringelly and Randwick. Each campaign measured hourly concentrations of isoprene and monoterpenes using the same PTR-MS instrument and employed standard calibration gases. Observations of monoterpenes by PTR-MS are based on the calibration and measurement of the combined monoterpene species at mass to charge ratio m/z = 81 for the Bringelly and Randwick campaigns and at mass to charge ratio m/z = 137 for the later SPS1, SPS2 and MUMBA campaigns. The change was made to improve sensitivity and reduce potential interferences. Three of the campaigns were documented in Emmerson et al. (2016): SPS1 and SPS2 were located at Westmead, a suburban site 21 km west of Sydney (150.9961°E, 33.8014°S). SPS1 ran from 18 February – 7 March 2011, and SPS2 from 14 April – 14 May 2012, (Cope et al., 2014). The Westmead site is located next to a grass playing field within hospital grounds, with a line of trees to the west and south, separating the site from trains, roads and housing beyond. The MODIS LAI value for Westmead is 1.2 m2 m-2. Dunne et al. (2018) have shown night time interference from wood smoke compounds in the isoprene signal taken during SPS2. Therefore the SPS2 isoprene observational dataset is restricted to daylight hours between 9am and 6pm. MUMBA was situated near the coast at Wollongong, (150.8995°E, 34.3972°S) from 22 December 2012 – 15 February 2013 (Paton-Walsh et al., 2017). The MUMBA site is also grassy (LAI of 1.7 m2 m-2), separated from the ocean 0.5 km to the east by a strip of eucalypt trees. A 400 m eucalypt forested escarpment is 3 km to the west.

A suite of meteorological data, including wind speed and direction were taken at each of the field campaign sites, with details given in the indicated literature. Polar bivariate plots are also shown in Figure 1 which give observed isoprene volume mixing ratios

by wind speed and direction at each of the campaign sites. These show the peak isoprene measurements and therefore the BVOC sources are not always associated with the dominant wind directions, but are correlated with the directions of the forested regions to the northwest and west of each of the sites.

2.1.1 Bringelly and Randwick

PTR-MS observations were undertaken in summer 2007 at Bringelly, a semi-rural site (150.7619°E, 33.9177°S, 24 January – 27 February 2007), and Randwick, 8 km from Sydney centre (151.2428°E, 33.9318°S, 28 February – 19 March 2007). Both sites are air quality management stations operated by the NSW government and take wind speed and direction, temperature and relative humidity measurements, along with ozone, NOx and particulate matter (www.environment.nsw.gov.au/AQMS/SiteSyd.htm). The inlet height for the PTR-MS instrument was approximately 4.5 m at both sites. Bringelly is located on reserve of open grassed council land (LAI of 2.1 m2 m-2), with occasional trees and bordered by Ramsay road at 53 m elevation. Low density housing is to the east. The heavily eucalypt forested Blue Mountains are 16 km to the west, which is where the source of the observed isoprene comes from. However the predominant wind directions are from the south-west and east.

The Randwick station at 28 m elevation is sited on a grassland paddock within army barracks, bordered by trees. The barracks are within a housing suburb (LAI of 0.5 m2 m-2). The dominant wind direction is from the south, with the dominant BVOC source coming from the north-west, consistent with the SPS1 BVOC source direction."

12. The authors use the term "emission factor" to mean several different things at various times during the analysis, in particular in Section 4 where they continue to use the same phrase to describe both a basal emission factor (i.e. emission rate at standard conditions) and a landscape emission factor (i.e. some kind of average gridcell BEF which accounts for contributions for all vegetation types within a model gridcell). I think

it is this phrasing that makes this section so hard to follow when the authors introduce the effect of differences in LAI between MEGAN and ABCGEM.

Response: We have restricted our use of emission rate to just be applicable to the isoprene and monoterpene measurements used to create the ABCGEM emission factors. We understand now that what we were referring to as an emission rate is actually an emission flux. We have standardised our use of emission flux, and opt to use the word 'emissions' in section 4.

13. Throughout: "inline" - do the authors mean "online"?

Response: Changed to be online.

Abstract:

14. L23: surely "simpler" rather than "simplified" as there is no suggestion that the authors have reduced ABCGEM in any way for this work.

Response: Changed to simpler

Introduction:

Throughout: there are far too many unsubstantiated statements made without reference to supporting literature, in particular:

15. L15: Please supply a reference for the C-CTM when first introduced.

Response: Done, the reference is Cope et al (2004). As this paragraph of the introduction has been re-written, we have ensured the first instance of C-CTM has this reference.

16. L21-22: Do the authors have evidence that these forests have a substantial impact?

Response: yes biogenic VOC emissions represent 55.3% of total non-methane VOC emissions for the Sydney Greater Metropolitan Region (GMR) (EPA, 2012).

However this paragraph of the introduction has been rewritten, and the above no longer

required.

17. L31-32: How is this relevant in a region where isoprene:monoterpene is unity?

Kanawade et al. 2011 suggests that the carbon ratio impacts the biogenic secondary organic aerosol formation. We were the first to observe carbon ratios of unity in our 2016 paper, but the impacts of this ratio in Australia has not been further studied.

Page 2 line 32 insert "however it is not known what impact a carbon ratio 1 will have"

This paragraph of the introduction has been re-written, so this line will now be included at page 8, line 27 as suggested by reviewer #1.

18. Methods: 2.1 As stated previously I would like to see far greater detail of the campaigns, the sites and meteorological conditions, the measurements available, etc. I consider this essential for the Brinsgelly and Randwick data which have not been previously published.

Response: This is dealt with in response to comment 11.

19. 2.2 As this model has not previously been described further detail is required in the main text. It would be far easier to follow if the authors presented the parameterisations here rather than attempt to describe in words.

Response: It is unfortunate that the details of ABCGEM were not published before, although it has been written up in a CSIRO report, and we include the reference Cope et al. (2009). However as we are not trying to validate ABCGEM, or make this into a model development paper, we include the details as supplementary material to avoid cluttering the main text. Reviewer #2 points out in comment #40 that ABCGEM is essentially based on parameterisations from Guenther et al, 1993 and 1997, so we feel it does not warrant inclusion in the main paper.

20. 2.2.1 Why set B to a value of 400 g m-2? This does not seem consistent with the description in Section 1 of the SI.

[Figure]

Response: The value of B = 400 g m-2 is just an example in order to put the emission rate value into the same units as those of the MEGAN emission factor maps. It represents an LAI of 4 m2 m-2 which is roughly the region where the majority of the eucalypt grid squares sit in our model domain. To avoid confusion, we have removed the example and have made reference to figure 2 here which gives the whole model domain of ABCGEM emission factors.

21. Also why devote so much of this section to a description of the grass PFT when it is promptly left out of the model?

Grass is not left out of the model. It is there, but provides a negligible contribution when compared with the emissions from eucalypt trees. In order to avoid confusion and make the paper flow better we will remove reference to the grass module of ABCGEM.

Page 3 line 26 insert "ABCGEM also accounts for grass emissions (see technical report by Cope et al. (2009)), however as eucalypt emissions dominate the Sydney air shed, the grass module will not be discussed here."

Other text relating to grass within the paper has been removed.

22. 2.2.2 This section is a discussion not a method. Further, as the authors devote so much time discussing the light-dependence of monoterpene emissions it would be useful to learn whether previous experiments on native Australian vegetation have shown evidence either way.

Response: Descriptions of the treatment of monoterpenes for each model have been moved to be included in each model's description in the methods section. The discussion and figure accompanying the section on activity functions has been moved to the supplementary section as recommended in question #45.

He et al (2000) made measurements of monoterpene emission rates from 15 eucalyptus species and found four of the strongest emitting species showed clear exponential temperature dependent relationships. There was no relationship with PAR.

Page 9 line 32 insert "This monoterpene relationship is consistent with He et al's (2000) study of 15 eucalypts in Australia, where they found four of the strongest emitting species showed strong exponential temperature dependent relationships, three with an r2 in excess of 0.9. While the range of PAR investigated was limited, He et al. (2000) found no relationship of eucalypt monoterpene emissions with PAR."

23. Section 3: Why is the C-CTM not described as part of the Methods section like L30: I suggest showing the model domains on Fig. 1.

Response: We have moved the model section up to be part of the methods section. The CSIRO CTM was section 3, which now becomes section 2.3 with 2 subsections. The rest of the paper has been renumbered accordingly.

3km model domains have been included in figure 1.

24. 3.1 I find this description of the seeming over dependence of ABCGEM on LAI extremely difficult to follow. However, if it is driven with 1980s LAI as a default it is good to see that the authors have conducted the AML sensitivity test.

Response: One of our major findings was that the result from ABCGEM did not change that much when a more up to date LAI product was used, leading us to conclude within the new section on temporal emission fluxes: "The AML domain average isoprene and monoterpene emission rates are 10% and 20% respectively different from ABCGEM and suggests that the choice (and age) of the LAI dataset is not critical to the BVOC emission estimates."

25. L25 Typo: "ccalculated"

Response: done

Section 4 Results and discussion

26. 4.1: As stated above I found this particularly hard to follow in part because of the use of "emission factor" to describe several things.

Response: see response 12

27. L14-23: I don't think that the "emission factors" being compared here are directly comparable. . .

Response: The MEGAN emission factor maps give values for e.g. isoprene for all vegetation types within a particular grid cell. The maps are in units of mg m-2 h-1 and are described in Guenther et al. (2006) as emission factors. What we have done is change the units of our standard condition emission rate from per gram of dry leaf to the same area based units as MEGAN by applying a landscape factor – in this case the column biomass.

In the choice of ABCGEM emission factors we include the sentence "These rates are converted into landscape emission factors for eucalypts by scaling with the column biomass of each grid cell, and are therefore a function of the LAI."

4.2 L8: See point below regarding Figure 4.

Response: figure 4 has been replaced with a temporal emissions plot.

28. L10-11: There would be no reason for anyone to expect the emissions to scale linearly with LAI . . . This comes back again to the confusing and inconsistent terminology used with regard to emission factors and emission rates.

Response: We have standardised our use of emission terminology as described in comment #12. Emissions at the leaf level are measured in units per gram of leaf. The larger the mass of leaves then the greater the emission. We have assumed that LAI (which is leaf area) is proportional with leaf mass. MEGANv2.1 also assumes scaling with LAI in that the array of emission activity of temperature independent species per layer (EatiLayer) is multiplied by LAI (canopy.f90 line 211).

Harrison et al. (2013) also find a linear relationship between isoprene emission and specific leaf area.

29. L20-21: The issue of whether or not monoterpene emissions should be treated as light dependent or not is really quite an important one as it is a fundamental mechanism rather than a model "artefact" such as LAI. Yet the authors seem to have made no attempt to investigate this further. To my mind this is where the real novelty could lie, and where the authors could offer something to the modelling community as a whole.

Response: We have taken this comment on board and made MEGAN-LDO our 4th sensitivity run for all the field campaigns. The results feature throughout the revised paper. See response to comment #4. We now believe the revised paper does have some novel results to offer the ACP community, and offers a suggestion for the application of MEGAN in south east Australia.

30. Is it the case that MEGAN is incorrect to assume that all monoterpene emissions have some light-dependence? Should these factors be PFT-specific? etc.

Response: No, some monoterpenes emitted in tropical regions do have a strong light dependence, but it is less so in temperate regions. There was only one light dependent factor assigned to each monoterpene species in MEGANv2.1, so it needed to be a global average. In MEGANv3 the light dependent factors will be driven by landscape specific parameters, so tropical regions can be different from temperate (personal communication from Alex Guenther, 25.10.17).

In the MEGAN model set-up section, add "In MEGAN all species, including monoterpenes, have a light dependency (Guenther et al., 2012), which were set using global average behaviours. Measurements of a-pinene fluxes in the tropics do show a light dependence (Rinne et al., 2002), whereas emissions from boreal pine forests and some eucalypts are described well using a temperature dependent function only (Tarvainen et al., 2005; He et al., 2000)."

31. L25-26: Again, surely this is to be expected . . .

Response: This section has now been replaced with the results from the temporal

emission plots.

32. Section 4.3 appears to be missing

Response: Thank-you, we have renumbered.

33. 4.4 L3-10: It would be helpful for the authors to include % differences to put their absolute changes into context.

Response: We have done this. This paragraph now reads

"In the isoprene difference plots, MEGAN predicts 1000 – 4000 g km-2 h-1 more isoprene to the west and north of Sydney than ABCGEM/AML, an increase of 40 – 200%. However MEGAN predicts 100 – 1000 g km-2 h-1 less isoprene than ABCGEM/AML in the urban regions where the field campaigns took place, contrary to the domain averages (at Westmead MEGAN is 15% lower, at Randwick, 46% lower). In this urban zone, MEGAN has a low fraction of plant coverage (30%) and an isoprene emission factor less than 3 mg m-2 h-1. In ABCGEM (and AML) the urban fraction of plant coverage and emission factors are dependent on the projected LAI which is 1 - 2 m2 m-2 here. Thus ABCGEM vegetation covers a larger area of the urban grid cells (39 - 63%), and the corresponding emission factor is also larger (2.8 - 5.7 mg m-2 h-1, or up to 47%) than MEGAN. These spatial patterns reiterate that a key difference between the two isoprene emission models is the input vegetation coverage."

34. 4.5 Overall I find the analyses of the output of the two model with respect to observations well done. However, for this paper to be suitable for ACP I feel that it is this section that should form the focus of the paper rather than the preceding consideration of LAI.

Response: We have taken on board these comments, and have removed the plot comparing ABCGEM and MEGAN emission fluxes with LAI to the supplementary material as advised. In order to concentrate on the temporal and spatial aspects of the emission rates we have plotted a time series of domain average emission rates from all the field

campaigns and included this as figure 4. The temporal plots show that each of the sensitivity runs captures the same synoptic features and the differences between them are essentially proportional.

35. p9, L3-6: Agreed. If updating a model developed 15 years ago to use a more up-to-date input dataset causes the model to "fail" the model clearly needs further development which is one of my key concerns with this work as it is presented. Why use this model rather than using the wealth of observations to improve the skill of MEGAN for this region? The two models take essentially the same approach to estimating emissions (i.e. empirical rather than mechanistic) so it is not evident what we gain from going back to the older model.

Response: We have rethought our stance on use of ABCGEM following comments from the two reviewers. We are not recommending use of ABCGEM in future, and will be concentrating on how best we can make improvements to the MEGAN description of Australian BVOC emissions. We have removed the paragraph in the conclusions section about improvements to ABCGEM.

We are not sure what the reviewer means by 'wealth of observations' as there have been very few experimental studies done on Australian vegetation, and almost none on vegetation in-situ.

36. L14-15: I feel the wind roses should be in the main paper as this consideration of emissions vs meteorology / chemistry is important.

Response: We have included observed isoprene polar bivariate plots in figure 1 to show the dominant wind directions, wind speeds and also the direction from which the highest isoprene volume mixing ratios are coming from. We have left the hourly wind roses presented in the supplementary material as there are too many of them to present in the main paper.

37. L20-25: MEGAN appears to have a good fit to the profile of monoterpene concentrations at Randwick and (to a lesser extent) SPS1 whereas at Bringelly all 3 simulations vastly over predict night-time concentrations. It seems far from clear that the light dependence of monoterpene emissions is the only issue here. I suggest the authors need to investigate more fully.

Response: The Bringelly monoterpene observations are very low compared to the other field sites, despite the close proximity of the eucalypt forests to the site. From the isoprene wind roses presented in figure 1 we can see that although the bulk of the biogenic material is coming from the west, the dominant wind direction is from the south-west and east. The wind direction is well predicted by the model.

We have investigated the issue further, and decided to include the analysis in the supplementary material. We have plotted observed and modelled wind roses for the Bringelly period, splitting them into daylight and night time hours. We have also plotted bivariate polar plots for the monoterpenes, despite them mainly existing at night when wind speeds are low. This can often give the false result that the source of the monoterpenes are local, but does indicate the source direction.

I agree that light dependence of monoterpenes is not the only issue at Bringelly.

Page 9 line 27 replace final sentence with "Light dependence is not the only issue at Bringelly, where the model is more influenced by stronger winds from the west and north than the observations, resulting in higher modelled BVOCs than observed. Further wind rose analysis is given in the supplementary material."

Supplementary material, section 7.1.

"The observed isoprene and monoterpene volume mixing ratios at Bringelly are lower than for other sites, despite the close (16km) proximity of the Blue Mountain region to the west. Additional wind roses are plotted for this field campaign, splitting the time period into daytime and night time (Figure 11). We also include wind roses for the modelled data, and also two polar bivariate plots for observed and modelled monoterpenes. During daytime, the observed and modelled wind direction is from the east, directly from the urban Sydney region. At night, when monoterpene levels are highest, the observed wind direction is from the south-south west, mostly at low wind speeds less than 2 m s-1. In the model, the direction of the peak monoterpenes has more of a south-westerly to westerly influence than the observations, at higher wind speeds up to 8 m s-1. We think the higher modelled wind speeds, and more westerly influence of the wind direction at night has contributed to the higher monoterpenes in the model. During the daytime when isoprene is more prevalent, the observed wind direction is away from the forests, keeping the observed isoprene low. In the model, there is a south westerly influence in the daytime with high wind speeds up to 10 m s-1, meaning the modelled isoprene is higher than observed."

38. P10, L1-5: I'm not sure that the q-q plots add much to the discussion and would suggest they be moved to the Supplementary.

Response: The q-q plots allow us to see how the modelled to observed bias changes as the volume mixing ratios increase. Now that the MEGAN-LDO test has been run we can see that the bias in isoprene and monoterpenes has improved considerably, particularly at the lower end of the concentration scale, for both species. The plot also shows us that additional research should look at the MEGAN biases for isoprene above observed values of 0.3 ppb, and for the MEGAN-LDO test at monoterpene levels greater than 1 ppb.

39. L19-24: See previous comments regarding the estimated uncertainty.

Response: see response 7

40. 5 Conclusions L25: I wouldn't consider the two models to have independent approaches to estimating emissions; both are based on the Guenther empirical algorithms developed in the 1990s.

Response: They are independent in that ABCGEM uses the LAI to scale the results

whereas MEGAN does not. We agree both are based on Guenther et al parameterisations and this is why ABCGEM is not suitable for a model development paper.

The first few sentences of the conclusions have been changed to read:

"The purpose of this work was to try and uncover reasons for the discrepancies produced by MEGAN in modelling BVOCs in the south east Australian region. This is a largely unstudied region with very few measurements of BVOC emissions. By making comparisons between locally developed ABCGEM and the well-established MEGAN model, both in terms of estimated emissions and also via simulated and observed atmospheric volume mixing ratios of isoprene and monoterpenes, we use local knowledge to suggest improvements for the application of MEGAN in Australia."

41. P11, L21-23: I would argue that in spite of using the same chemistry scheme some of the difference in concentrations between the two models will still be due to chemical processing as it is highly non-linear and strongly dependent on VOC-Nox ratios which will differ if VOC emissions change. I agree that the predominant cause of the differences are differences in emissions.

Response: We did observe changes in isoprene at MUMBA (the hottest campaign) in the MEGAN-LDO test due to the 163% increase in monoterpenes, which affected the oxidant volume mixing ratios by 0.1 to a few ppt.

We have added the following to the first paragraph of section 3.4:"The transport and chemical schemes are the same in each model therefore for any particular campaign, the bulk of the differences between the ABCGEM and MEGAN models should directly scale to the differences in emissions between the models"

We have added an additional paragraph to the section 3.4 "Changes to the oxidants as a result of the additional monoterpenes in the MEGAN-LDO test has impacted on the isoprene at the campaign sites, in general reducing MEGAN daytime isoprene by 4% and night time isoprene by 15%. MEGAN-LDO has also improved the percentage of

points within a factor of 2 of the observations for isoprene. This is not the case for iso-prene at MUMBA which has increased during the daytime by 55% and at night by 18%, reducing the percentage within a factor of 2 of the observations to 4%. This is because the monoterpene levels in the MEGAN-LDO test have increased by 163% at night and 65% during the day over the very hot January 2013 of the MUMBA campaign, more than for any other field campaign, impacting the oxidant chemistry. Peak modelled OH for MUMBA has decreased by 0.1 ppt ($\sim$1700%) and HO2 by 1.5 ppt ($\sim$350%)."

42. L31: As noted previously I feel the authors need to give far more detail of these two sets of measurements as they have not been previously published.

Response: This has been done. See response 11

43. p12, L7-8: Agreed, but that doesn't preclude the authors from investigating further at this stage.

Response: See response 35. We will remove this paragraph about improvements to ABCGEM.

44. Table 2. What about PAR and Temperature for each campaign? (average and some measure of range)

Response: There are PAR measurements only for SPS2 and MUMBA which were pre-sented in Emmerson et al 2016. We have added the observed and modelled average and range of temperatures for each campaign to Table 2.

45. Figures 2 -4 would be better presented in the SI (but see below) Figures 3 & 4 I feel the way the data is presented is fundamentally flawed. LAI is not a discrete variable but rather a weighted average for a grid cell based on proportional land cover. It therefore makes no sense to plot the data showing ranges for emissions but not LAI. While I understand that the authors have binned the data by a range of LAI it is still not appropriate to plot the emission rate against the mid-point of the LAI bin. At the very least, it should be a weighted average of the LAIs of the grid cells within that bin but

even then I would question its appropriateness.

We have adjusted all the figures where LAI is used on the x-axis to be the weighted average of the LAIs within each bin, where the weighting refers to fraction of land occupied by each LAI bin. In ABCGEM the fraction of land covered by vegetation is related to LAI.

We have removed figure 4 and the associated text in section 4.2 to the supplementary section. However we still feel the comparison of ABCGEM and MEGAN emission factors is important and will keep figure 3 within the main paper.

Page 5 line 33 insert "Here LAI is weighted by the fractional area taken up by each bin."

References

Cope, M., Keywood, M., Emmerson, K., Galbally, I., Boast, K., Chambers, S., Cheng, M., Crumeyrolle, S., Dunne, E., Fedele, F., Gillett, R. W., Griffiths, A., Harnwell, J., Katzfey, J., Hess, D., Lawson, S., Miljevic, B., Molloy, S., Powell, J., Reisen, F., Ristovski, Z., Selleck, P., Ward, J., Zhang, C., and Seng, J.: The Sydney Particle Study. CSIRO, Australia. Available at http://www.environment.nsw.gov.au/aqms/sydparticlestudy.htm, 2014.

Cope, M. E., Lee, S., Noonan, J., Lilley, B., Hess, D., and Azzi, M.: Chemical transport model: Technical description, CSIRO Marine and Atmospheric Research Internal Report 2009.

Dunne, E., Galbally, I. E., Cheng, M., Selleck, P., Molloy, S. B., and Lawson, S. J.: Comparison of VOC measurements made by PTR-MS, Adsorbent Tube/GC-FID-MS and DNPH-derivatization/HPLC during the Sydney Particle Study, 2012: a contribution to the assessment of uncertainty in current atmospheric VOC measurements, Atmos. Meas. Tech., 11, 141-159, https://doi.org/10.5194/amt-11-141-2018, 2018.

Emmerson, K. M., Galbally, I. E., Guenther, A. B., Paton-Walsh, C., Guerette, E. A., Cope, M. E., Keywood, M. D., Lawson, S. J., Molloy, S. B., Dunne, E., Thatcher, M.,

Karl, T., and Maleknia, S. D.: Current estimates of biogenic emissions from eucalypts uncertain for southeast Australia, Atmos Chem Phys, 16, 6997-7011, 10.5194/acp-16-6997-2016, 2016.

Guenther, A. B., Jiang, X., Heald, C. L., Sakulyanontvittaya, T., Duhl, T., Emmons, L. K., and Wang, X.: The Model of Emissions of Gases and Aerosols from Nature version 2.1 (MEGAN2.1): an extended and updated framework for modeling biogenic emissions, Geosci Model Dev, 5, 1471-1492, DOI 10.5194/gmd-5-1471-2012, 2012.

Harrison, S. P., Morfopoulos, C., Dani, K. G. S., Prentice, I. C., Arneth, A., Atwell, B. J., Barkley, M. P., Leishman, M. R., Loreto, F., Medlyn, B. E., Niinemets, U., Possell, M., Penuelas, J., and Wright, I. J.: Volatile isoprenoid emissions from plastid to planet, New Phytol, 197, 49-57, 10.1111/nph.12021, 2013.

He, C. R., Murray, F., and Lyons, T.: Monoterpene and isoprene emissions from 15 Eucalyptus species in Australia, Atmos Environ, 34, 645-655, Doi 10.1016/S1352-2310(99)00219-8, 2000.

Orians, G. H., and Milewski, A. V.: Ecology of Australia: the effects of nutrient-poor soils and intense fires, Biol Rev, 82, 393-423, 10.1111/j.1469-185X.2007.00017.x, 2007.

Rinne, H. J. I., Guenther, A. B., Greenberg, J. P., and Harley, P. C.: Isoprene and monoterpene fluxes measured above Amazonian rainforest and their dependence on light and temperature, Atmos Environ, 36, 2421-2426, Pii S1352-2310(01)00523-4 Doi 10.1016/S1352-2310(01)00523-4, 2002.

Tarvainen, V., Hakola, H., Hellen, H., Back, J., Hari, P., and Kulmala, M.: Temperature and light dependence of the VOC emissions of Scots pine, Atmos Chem Phys, 5, 989-998, 2005.

---

## Author Comment (AC3) · 14 Feb 2018

Many air quality modellers use the word concentration in a generic sense as applied to volume mixing ratios of substances, e.g. a concentration of 4 ppb isoprene.

However we find this can be misleading or ambiguous as the specific definition of 'concentration' is a mass of a substance per unit volume.

Therefore we have replaced the word concentration with either the term 'volume mixing ratio', or 'level'. This means a change to the paper's title, now becoming:

[Figure]

"Isoprene and monoterpene emissions in south east Australia: comparison of a multi-layer canopy model with MEGAN and with atmospheric observations"

---

## Author Response (AR2)

Dear Editor,

Thank-you for your consideration of our manuscript.

In the last iteration of the manuscript we made a small change to the manuscript title. We hope this can be incorporated when the manuscript is published in ACP?

5 Kathryn Emmerson

April 2018

Response to reviewer #1

In one last proofreading, there are only a few minor text corrections to address, for instance remove parenthesis in

10 line 18 in "(BVOCs)".

Thank-you. I have corrected the mistake found by reviewer #1 and completed a full proofread of the manuscript.

P 5 line 25. Add "147" to this line and remove the following sentence "MEGAN predicts the emission rates of 147 BVOCs."
P11 line 11. Replaced 'utilizing' for 'using'
15 P11 line 28 Added 'used'
P12 line 2 Sentence beginning MEGAN, insert "monoterpene" between MEGAN and emissions.
Acknowledgements. Added 'NSW Office of Environment and Heritage'

Response to reviewer #2.

20 I thank the authors for their thorough response to my (and the first reviewer's) comments. The textual changes have substantially improved the clarity of the methodology, results and conclusions. While I still have reservations with regard to their justification for the use of ABCGEM study and their over-emphasis of the importance of their conclusions regarding land cover, I am satisfied that the manuscript is now suitable for publication in ACPD.

Specific comments:

I am particularly pleased that the authors have conducted a sensitivity test on the light-dependency of monoterpene emissions from Australian eucalypt ecosystems. The findings of this additional test are well-presented, well-explained and, to my mind, reach a conclusions that does mark a clear step forward in emissions

30 modelling. Importantly this is a finding that is globally applicable and relevant.

Likewise, the extended uncertainty analysis demonstrating the consistency between top-down and bottom-up estimates of the uncertainty in emission rates makes a valuable contribution. As the authors point out, what is now needed is a long-term comprehensive series of atmospheric and leaf-level measurements to further constrain

35 uncertainties.

I thank the authors for the extended description of the various measurement sites and campaigns. I find this makes the comparisons and model evaluations much more meaningful and far easier to follow.

40

I still feel that the authors place far too much emphasis on their conclusion that modelled landscape scale

emissions are critically dependent on the landcover maps used to drive the mode. Regardless of whether or not it is the first time this has been published for Australia it is NOT a novel finding. There is a canon of literature from all world regions (e.g. Arneth et al. (2011; global); Langford et al. (2010; SE Asia), Warneke et al. (2010; US), Huang et al. (2015; US), Zhao et al. (2016; US), Otter et al., 2003; Africa)) to demonstrate precisely that.

We have included these references in the discussion where appropriate.

Page 6 line 23. "similar to the findings of Arneth et al. (2011), Zhao et al. (2016), Huang et al. (2015), Otter et al. (2003), Warneke et al. (2010) and Langford et al. (2010)."

Page 8 line 3, added 'again' to "This again is an illustration of the deficiencies in vegetation mapping adversely affecting BVOC emissions modelling".

In the conclusions, we leave the last comment in the last paragraph: "We have highlighted the roles of the spatial and temporal distributions of LAI and the correct mapping of plant species or plant functional types in this modelling".

But remove the sentence from the first paragraph of page12: "this is a practical example of the impact of differences in input data on BVOC emissions modelling."

The authors state that they have chosen to confront the MEGAN model with ABCGEM because MEGAN in its "default" setting proved to perform poorly over Australia. Emmerson et al. (2016) concluded that the reason for that poor fit was the use of incorrect emission factors but the authors now seem to be arguing that in fact it is a more fundamental problem: that Australia's unique isolation and environmental conditions have resulted in the evolution of different emissions mechanisms. There is certainly a wealth of evidence that VOC synthesis has evolved independently in numerous lines of vegetation at different times. However, the emissions algorithms in ABCGEM are based on the original Guenther et al (1993) empirical formulations, as are those in MEGAN. This study can not therefore be claimed to be investigating any such fundamental difference in VOC synthesis. The authors need to make this entirely clear.

We will add the following to page 2 line 20:

 "These questions on VOC synthesis are beyond the scope of this paper. Simpler causes of model-observation mismatch are explored first."

Finally, I still find the authors to be muddled in their description of emission factors. Until the emission factors in ABCGEM have been scaled up to the landscape scale they are NOT comparable with the gridded emission factors that are used with MEGAN. However, overall I find the discussion of emission factors and emissions much clearer and thank the authors for clarifying their terminology.

The biomass term in equation 1 of the supplementary section, with units of g m$^{-2}$ is per area of ground, not per area of leaf, which is where we think reviewer #2 is misunderstanding our definition? (Bm is the total dry weight of leaves extending from the ground to the canopy/grass height, hC per unit area of ground). Therefore the ABCGEM emission factors are scaled up to the landscape scale. To clarify this we change page 4 at line 5: "We take measured leaf level emission rates and convert them into landscape emission factors for eucalypts by scaling with the column biomass of each grid cell (per unit ground area), making them a function of the LAI (see supplementary material equation 3)."

We also change page 5, line 3, changing "the column biomass per unit ground area, $B_m$".

Overall, I now feel that this is a robust and substantial study and strongly recommend publication in its present form.

We thank reviewer #2 for their comments.

[revised manuscript text omitted]